# NeurIPT: Foundation Model for Neural Interfaces

**Zitao Fang**[1]    **Chenxuan Li**[1]    **Hongting Zhou**[1]    **Shuyang Yu**[2]    **Guodong Du**[3]
**Ashwaq Qasem**[1]    **Yang Lu**[4]    **Jing Li**[5]    **Junsong Zhang**[4*]    **Sim Kuan Goh**[1*]
[1]Xiamen University Malaysia    [2]Columbia University
[3]The Hong Kong Polytechnic University    [4]Xiamen University
[5]Harbin Institute of Technology (Shenzhen)
ait2209071@xmu.edu.my    zhangjs@xmu.edu.cn    simkuangoh@gmail.com

## Abstract

Electroencephalography (EEG) has wide-ranging applications, from clinical diagnosis to brain-computer interfaces (BCIs). With the increasing volume and variety of EEG data, there has been growing interest in establishing foundation models (FMs) to scale up and generalize neural decoding. Despite showing early potential, applying FMs to EEG remains challenging due to substantial inter-subject, inter-task, and inter-condition variability, as well as diverse electrode configurations across recording setups. To tackle these open challenges, we propose **NEURIPT**, a foundation model developed for diverse EEG-based **Neur**al **I**nterfaces with a **P**re-trained **T**ransformer by capturing both homogeneous and heterogeneous spatio-temporal characteristics inherent in EEG signals. Temporally, we introduce Amplitude-Aware Masked Pretraining (AAMP), masking based on signal amplitude rather than random intervals, to learn robust representations across varying signal intensities beyond local interpolation. Moreover, this temporal representation is enhanced by a Progressive Mixture-of-Experts (PMoE) architecture, where specialized expert subnetworks are progressively introduced at deeper layers, adapting effectively to the diverse temporal characteristics of EEG signals. Spatially, NeurIPT leverages the 3D physical coordinates of electrodes, enabling effective transfer of embedding across varying EEG settings, and develops Intra-Inter Lobe Pooling (IILP) during fine-tuning to efficiently exploit regional brain features. Empirical evaluations across eight downstream BCI datasets, via fine-tuning, demonstrated NeurIPT consistently achieved state-of-the-art performance, highlighting its broad applicability and robust generalization. Our work pushes forward the state of FMs in EEG and offers insights into scalable and generalizable neural information processing systems. Our project is available at this https URL.

## 1   Introduction

Electroencephalography (EEG) has been widely adopted as a proxy for brain activity and dynamics, due to its non-invasiveness, portability, and high temporal resolution for real-time monitoring [1]. EEG facilitates investigations into brain function, assists in neurological diagnoses [2], provides objective biomarkers for cognitive and affective states [3], and enables the development of brain-computer interfaces (BCIs) [4]. The growing availability of large-scale EEG datasets (spanning diverse populations, recording configurations, and experimental paradigms) has spurred the development of a wide range of computational approaches, including machine learning and deep learning models (CNN [5], RNN [6], GNN [7], Transformers [8]) for learning representations. However, these models are often tailored to specific tasks and settings, limiting their generalizability and cross-setting applicability. Hence, a paradigm shift is necessary to establish generalizable models that can effectively leverage

---

*Corresponding authors.

39th Conference on Neural Information Processing Systems (NeurIPS 2025).

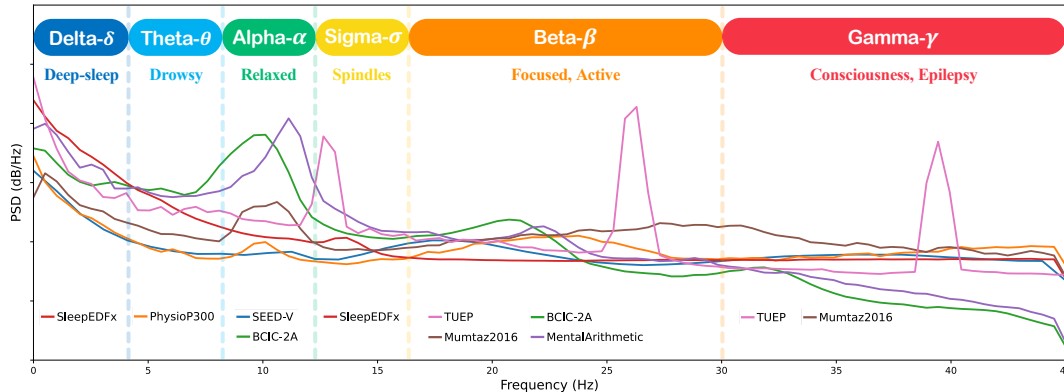

Figure 1: Spectrograms of the nine EEG datasets reveal both homogeneous and heterogeneous spectral patterns, with some datasets showing higher power spectral density (PSD) in specific EEG frequency bands. Thus, they demand neural representations capable of adapting to input variability.

EEG signals across diverse setups and settings, including inter-subject variability, differences between healthy and patients, electrode configurations, and variations in cognitive or behavioral tasks.

Recent advances in foundation models (FMs), large-scale neural architectures pre-trained using transformers on diverse and unlabeled datasets through self-supervised learning [9], followed by fine-tuning on downstream datasets, have demonstrated remarkable success in natural language processing (NLP) and computer vision (CV). In NLP, models [10] such as Claude, DeepSeek, Gemini, GPT, and Llama, pre-trained on massive text corpora, have learned general-purpose representations that transfer effectively to a wide range of downstream tasks [11]. Similarly, pre-trained Vision Transformers (ViTs) [12], Masked Autoencoder (MAE) [13] have become foundational components in modern CV pipelines. These developments highlight the potential of FMs to produce robust, transferable representations of both images and language. Building on this progress, multimodal large language models (MLLMs) (e.g., SORA [14]) further extend these capabilities by integrating information across multiple data modalities, such as text, images, and audio. Hence, it is intriguing to investigate FMs' generalizability to brain signals by developing EEG-based FMs.

Initial efforts to develop foundation models (FMs) for neural decoding showed early promise. BENDR [15] adapted techniques from language modeling to learn compressed representations of raw data signals, enabling the model to generalize across different tasks and datasets. EEG2Vec [16] learned generative-discriminative representations for emotion classification tasks. BIOT [17] addressed the challenges of cross-dataset EEG learning by tokenizing EEG signals into fixed-length segments, accommodating variable-length sequences and mismatched channels. By learning from diverse biosignal datasets, improved performance can be achieved in tasks such as seizure detection. To encode EEG signals into discrete tokens, LaBraM [18] introduced a neural tokenizer with vector-quantized spectrum prediction, facilitating the pretraining of transformers to predict masked segments and enhancing the model's ability to learn from large-scale EEG data. NeuroLM [19] proposed a universal multi-task foundation model that bridged the gap between language and EEG signals. Treating EEG as a foreign language, NeuroLM adopted a text-aligned neural tokenizer and leveraged large language models (LLMs) to perform multi-task learning across various EEG-

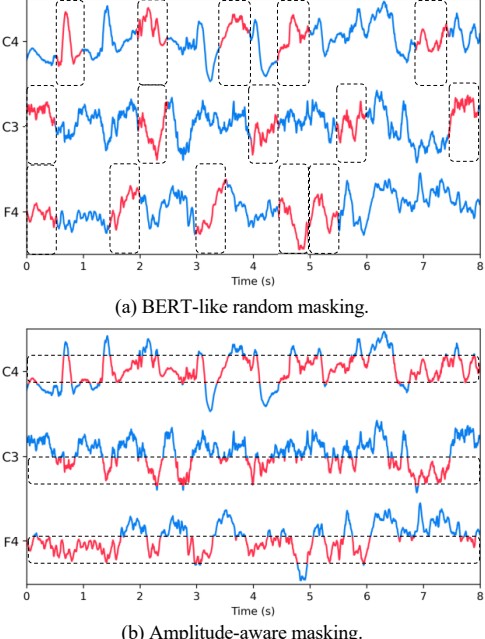

(a) BERT-like random masking.

(b) Amplitude-aware masking.

Figure 2: BERT-like random masking v.s. our proposed amplitude-aware masking.

related tasks. EEGPT [20] introduced a dual self-supervised learning approach for feature alignment, capturing spatio-temporal representations in EEG data. This method enhanced the model's ability to learn robust features that generalized across different tasks and datasets. CBraMod [21] presented a criss-cross transformer architecture designed to model spatial and temporal dependencies separately in EEG signals from diverse datasets, facilitating linear probing for fine-tuning on numerous downstream tasks. These methods provided evidence of the applicability of FMs for EEG.

Despite the encouraging preliminary findings, current foundation model (FM) methods for EEG have largely adopted pretraining strategies from language and time-series domains without accounting for several unique properties of EEG. First, existing positional encodings treat electrode channels as interchangeable and ignore their physical three-dimensional arrangement, losing critical spatial relationships, which can significantly impair transferability. Second, prevalent masked pretraining strategies are typically based on randomly masking contiguous signal segments like BERT [22], unintentionally guiding the model toward local interpolation rather than meaningful global representation learning, as illustrated in Figure 2. Third, conventional neural architectures rely heavily on fully connected layers or global pooling mechanisms when fine-tuned to downstream tasks and thus do not explicitly utilize regional brain features. Finally, due to the diversity and complexity of EEG data patterns, ranging from slow-wave oscillations during sleep to rapid spikes during seizures [23], models must be capable of adaptively capturing heterogeneous temporal dynamics recorded in diverse setups. The diverse spectral patterns are shown in Figure 1. Addressing these challenges is critical to fully realizing the potential of foundation models for EEG.

To address the aforementioned challenges, we introduce NEURIPT, a novel EEG foundation model developed to learn robust and generalizable representations across diverse BCI applications. Our NEURIPT include: (i) 3D-Aligned Spatial Encoding, a flexible positional encoding scheme that integrates the actual three-dimensional coordinates of EEG electrodes, enabling seamless adaptation across varying electrode montages without re-training; (ii) Amplitude-Aware Masked Pretraining (AAMP), a novel masking strategy guided by EEG signal amplitude rather than random intervals, shown in Figure 2, compelling the model to capture underlying EEG patterns with amplitude serves as a proxy for signal energy instead of trivial local interpolations; (iii) Progressive Mixture-of-Experts (PMoE), an architectural innovation wherein the number of specialized subnetworks increases with model depth, effectively accommodating diverse temporal EEG patterns; and (iv) Intra-Inter Lobe Pooling (IILP), a spatial aggregation method that performs hierarchical pooling within and across brain lobes, leveraging distinctive regional brain features for downstream tasks. Empirically, NEURIPT achieves consistent state-of-the-art performance across eight diverse EEG benchmarks, including seizure detection, cognitive state decoding, sleep stage classification etc., summarized in Figure 5. Our comprehensive evaluations highlight NEURIPT 's enhanced scalability, improved robustness to spatial and temporal variations, and progress toward building truly universal EEG representation models. These findings mark an advancement for foundation models in EEG and offer practical insights for future neural interface development.

The main contributions of this paper are summarized as follows:

1. We propose NEURIPT, a foundation model developed for EEG-based neural interfaces, designed to learn robust and generalizable representations by capturing the intrinsic spatio-temporal heterogeneity of EEG signals across diverse settings.

2. Temporally, we introduce Amplitude-Aware Masked Pretraining (AAMP), which masks segments based on signal amplitude rather than random intervals, enabling the model to learn robust features across varying signal intensities. In addition, we design a Progressive Mixture-of-Experts (PMoE) architecture that adaptively introduces specialized experts at deeper layers to capture diverse temporal dynamics in EEG.

3. Spatially, NEURIPT incorporates the 3D physical coordinates of electrodes for embedding, facilitating spatial generalization across datasets with varying sensor montages. During fine-tuning, we introduce Intra-Inter Lobe Pooling (IILP) to explicitly model and leverage region-specific brain activity patterns for downstream tasks.

4. We conducted extensive empirical evaluations on eight benchmark BCI datasets. NEURIPT consistently achieved state-of-the-art performance, demonstrating strong generalization and broad applicability across diverse EEG-based tasks.

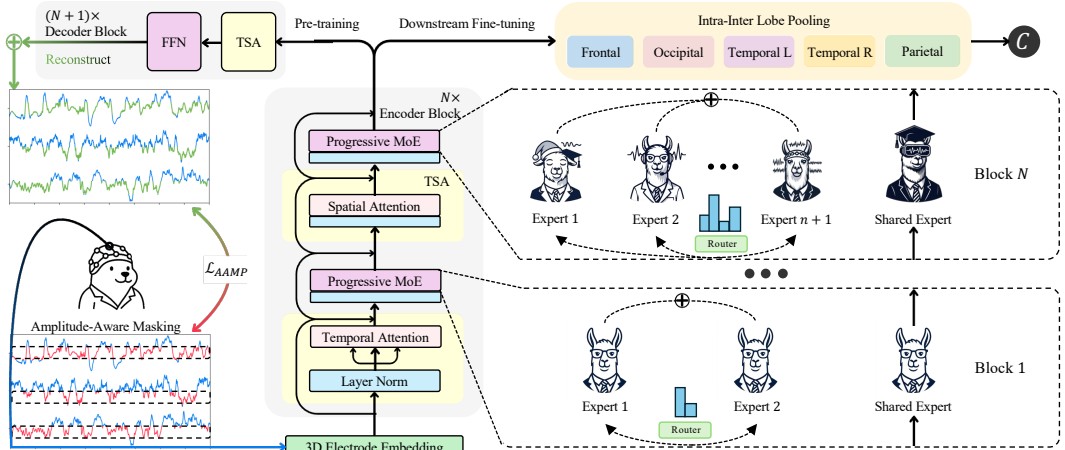

Figure 3: Overview of our **NEURIPT**, which comprises Amplitude-Aware Masked Pretraining (AAMP), 3D Electrode Embedding, Progressive Mixture-of-Experts (PMoE), and Intra-Inter Lobe Pooling (IILP) for fine-tuning. See Figure 4 for details on the IILP module.

## 2 Method

The framework of NEURIPT is illustrated in Figure 3. In Section 2.1, we provided the details on how NEURIPT integrates spatial and temporal EEG representations by embedding 3D spatial electrode coordinates and employing Amplitude-Aware Masking to enhance self-supervised learning. Subsequently, Section 2.2 describes our Progressive Mixture-of-Experts architecture to dynamically manage EEG variations, complemented by Intra-Inter Lobe Pooling to effectively capture regional brain features for downstream tasks.

### 2.1 Embedding Spatial Context into Temporal Representations

We pretrain our NEURIPT using self-supervised learning to learn robust representations of EEG signals across diverse setups and settings, without relying on annotations or incorporating any downstream EEG task data. Specifically, given a set of unannotated EEG dataset $\mathcal{D}^{(u)} = \left\{\mathbf{x}^{(i)}\right\}_{i=1}^{N}$, where each sample $\mathbf{x}^{(i)} \in \mathbb{R}^{T \times D}$ represents multivariate EEG signals with $T$ time steps and $D$ electrode channels.

**3D Electrode Embedding** This is designed to leverage the three-dimensional spatial relationships among EEG electrodes, which are typically arranged according to the international 10–20 electrode placement system. Specifically, for the $d$-th electrode channel positioned at $(x_d, y_d, z_d)$, we encode its three spatial coordinates separately and concatenate them as follows:

$$PE_d^{(s)} = \text{Concat}\big(PE_x\left(x_d\right), PE_y\left(y_d\right), PE_z\left(z_d\right)\big). \tag{1}$$

Each coordinate embedding uses a sinusoidal function defined as $PE_\alpha(pos)_{2j} = \sin(pos \cdot 10000^{-2j/d_\alpha})$ and $PE_\alpha(pos)_{2j+1} = \cos(pos \cdot 10000^{-2j/d_\alpha})$, where $\alpha \in \{x, y, z\}$ denotes the spatial axes, $pos$ represents the position along the corresponding spatial coordinate, $j$ indexes the embedding dimensions, and $d_\alpha = d_{model}/3$ is the dimensionality of the embedding for axis $\alpha$.

Building upon pixel-level embeddings in CV [12], we adopt single-point embeddings to preserve temporal details and effectively capture sharp spikes crucial for brain activity analysis. We represent each EEG sample as a collection of data points $\mathbf{x}^{(i)} = \{x_{t,d}^{(i)} \mid 1 \le t \le T, 1 \le d \le D\}$, each point is then embedded with both temporal and spatial information:

$$\mathbf{s}_{t,d}^{(i)} = \mathbf{E}\mathbf{x}^{(i)} + PE^{(t)} + PE^{(s)}. \tag{2}$$

Here, $\mathbf{E} \in \mathbb{R}^{d_{model}}$ denotes a learnable linear projection vector, $PE^{(t)}$ represents temporal positional encoding following the sinusoidal formulation from Vaswani et al. [9]. After embedding, the encoder

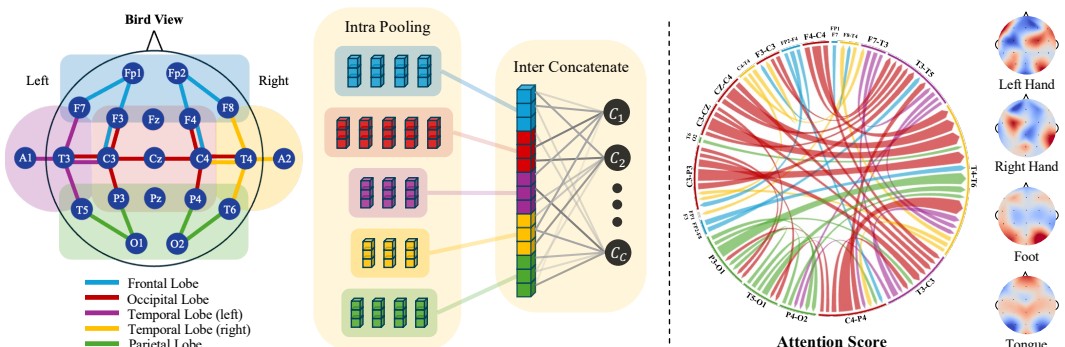

Figure 4: (Left) Intra-Inter Lobe Pooling (IILP) leverages regional brain features during fine-tuning. (Right) Visualization of attention scores from the temporal attention module and analysis of Pearson correlation between class logits and channel perturbation using Gaussian multiplicative noise. Note that colors in the right panel correspond to the brain regions depicted on the left.

input is represented as follows:

$$\mathbf{S}^{enc} = \left\{ \mathbf{s}_{t,d}^{(i)} \middle| t = 1, \ldots, T, d = 1, \ldots, D \right\}, \tag{3}$$

where each vector $\mathbf{s}_{t,d}^{(i)}$ encapsulates both spatial and temporal features, which are crucial for capturing EEG dynamics in subsequent modeling stages. Both temporal and spatial encodings natively support varying temporal lengths $T$ and spatial dimensions $D$, enabling seamless adaptation to diverse EEG electrode placement systems, such as the 10-05 and 10-20 standards, without additional convolutional components or padding as used in EEGPT [20] and CBraMod [21]. This design is also naturally consistent with the proposed AAMP method introduced below.

**Amplitude-Aware Masked Pretraining (AAMP)** Self-supervised learning techniques like BERT [22] and MAE [13], which reconstruct randomly masked data, have shown promising results in multivariate time series (MTS) [15, 18, 20]. However, random masking often degrades to simple interpolation between unmasked points in time series data [24], thereby limiting the extraction of meaningful structural representations. To address this, we propose a novel Amplitude-Aware Masking Pretraining (AAMP) approach, explicitly designed to capture more informative features while avoiding mere interpolation, illustrated in Figure 2. After obtaining the embedding array $\mathbf{S}$, we generate our Amplitude-Aware Masking:

$$\mathcal{M} = \left\{ \mathbb{1} \left\{ x_{t,d}^{(i)} \in [\mathcal{L}_d, \mathcal{U}_d] \right\} \mid t = 1, \ldots, T, d = 1, \ldots, D \right\}, \tag{4}$$

where the interval $[\mathcal{L}_d, \mathcal{U}_d]$ covers $T \cdot \mathcal{P}$ points, centered around the $\xi_d^{(i)} \cdot T$-th point in $sorted(\mathbf{x}_d^{(i)})$. Specifically, probability $\mathcal{P}$ denotes the masking ratio and $\xi_d^{(i)} \sim \mathcal{U}(0, 1)$ is a randomly sampled percentile for dimension $d$ of data instance $i$. Subsequently, we derive the EEG embedding $\mathbf{S}$, partitioned into the masked set $\mathbf{S}^{\mathcal{M}} = \{\mathbf{s}_{t,d} | m_{t,d} = 1\}$ and the unmasked set $\mathbf{S}^{\mathcal{U}} = \{\mathbf{s}_{t,d} | m_{t,d} = 0\}$ according to the corresponding $\mathbf{s}_{t,d}$ and $m_{t,d}$.

**Hierarchical Attention Modules** In complex MTS tasks, i.e., EEG signals processing, capturing both temporal and spatial dimensions is significant [21, 25]. To effectively extract these spatio-temporal characteristics at multiple granularities, we adopt Crossformer [26] as part of our backbone architecture, which hierarchically captures alternating temporal and spatial dependencies by using `TSA` module. Further implementation details of our modified Crossformer are provided in Appendix C.1.

Specifically, our hierarchical attention model comprises an encoder $\mathbb{ENC}_{\theta_E}$ and a decoder $\mathbb{DEC}_{\theta_D}$, each equipped with trainable parameters $\theta_E$ and $\theta_D$, respectively. These components jointly capture spatio-temporal relationships and reconstruct masked portions of the input data as follows:

$$\hat{\mathbf{x}} = \mathbb{DEC}_{\theta_d} \left( \mathbb{ENC}_{\theta_e} \left( \mathbf{S}^{enc} \right), \mathbf{S}^{dec} \right), \tag{5}$$

and the decoder input $\mathbf{S}^{dec}$ is formulated as:

$$\mathbf{S}^{dec} = \mathbf{E} \left( (1 - \mathcal{M}) \odot \mathbf{x}^{(i)} \right) + PE_t^{(t)} + PE_d^{(s)}. \tag{6}$$

The encoder $\mathbb{ENC}_{\theta_e}$ processes the unmasked input $\mathbf{S}^{\mathcal{U}}$ to extract meaningful features while setting the attention scores of the masked set $\mathbf{S}^{\mathcal{M}}$ to zero, thereby preventing unintended access to masked tokens. Thereafter, the decoder $\mathbb{DEC}_{\theta_d}$ reconstructs the signals in $\mathbf{S}^{dec}$ through a specific [mask] token, leveraging both the encoder outputs and unmasked signals enriched by comprehensive positional information. We then optimize the parameters $\theta_E$ and $\theta_D$ by minimizing the $\ell_p$-norm reconstruction loss:

$$\mathcal{L}_{\text{AAMP}}(\theta_E, \theta_D) = \frac{1}{n}\left(\sum_{i=1}^{n}\|\mathbf{x}^{(i)} - \hat{\mathbf{x}}^{(i)}\|^p\right)^{1/p}. \tag{7}$$

## 2.2 Integrating Temporal Dynamics into Spatial Representations

EEG signals exhibit complex, heterogeneous information across various frequency bands, transient events, and even inevitable artifacts, making their representations challenging for a single feed-forward network (FFN). To address this, we leverage a Progressive Mixture-of-Experts (PMoE) architecture, enabling distinct sub-networks as specialized experts to capture specific EEG features, combined with a shared expert to ensure stable generalization. Subsequently, we fine-tune with Intra-Inter Lobe Pooling (IILP), which effectively captures cross-channel connectivity patterns.

**Progressive Mixture-of-Experts (PMoE)**   Given $\hat{\mathbf{Z}}^l \in \mathbb{R}^{T \times D \times d_{model}}$, the output after the attention layer and subsequent layer normalization, a gating mechanism computes token-level routing weights:

$$g^l = \textit{TopKSoftmax}(\text{Router}^{(l)}(\hat{\mathbf{Z}}^l)), \quad \sum_{e=1}^{E_l} g_e^l = 1, \tag{8}$$

where $E_l$ denotes number of experts at the $l$-th encoder layer and each expert $e$ applies its private sub-network $Y_e^l = \text{FFN}_e^{(l)}(\hat{\mathbf{Z}}^l)$. Introduced by [27], *TopKSoftmax* applies the *softmax* function exclusively to the top-$k$ logits, representing the sparse activation that serves to save computation. These representations are aggregated through gating weights $g$ and further produced the output of transformer block $l$:

$$\text{PMoE}^{(l)}(\hat{\mathbf{Z}}^l) = \sum_{e=1}^{E_l} g_e^l \odot Y_e^l + \text{FFN}_{shared}^{(l)}(\hat{\mathbf{Z}}^l), \tag{9}$$

$$\mathbf{Z}^l = \text{PMoE}^{(l)}(\hat{\mathbf{Z}}^l) + \tilde{\mathbf{Z}}^l, \tag{10}$$

where $\odot$ denotes element-wise multiplication, $\text{FFN}_{shared}$ denotes the shared expert, and $\tilde{\mathbf{Z}}^l$ denotes the residual connection from the attention output. Then we incorporate an auxiliary loss $\mathcal{L}_{\text{aux}}$ to encourage balanced expert utilization and robust routing behavior [27]. Please refer to Appendix C.1 and C.3 for more details about specific architecture and MoE, respectively.

The shared expert captures generalizable patterns, while the progressively introduced experts handle increasingly specialized signal features [28], effectively adapting to the inherent complexity and variability of EEG signals.

**Finetuning with Intra-Inter Lobe Pooling (IILP)**   Following the pretraining phase, we performed fine-tuning using downstream labeled datasets $\mathcal{D}^{(l)} = \{(\mathbf{x}^{(j)}, \mathbf{y}^{(j)})\}_{j=1}^{n}$ comprising $n$ instances. Each instance $\mathbf{x}^{(j)} \in \mathbb{R}^{T \times D}$ is associated with a class label $\mathbf{y}^{(j)} \in \mathcal{C} := \{1, \ldots, C\}$, indicating the downstream classification category.

To better capture EEG functional connectivity patterns while suppressing redundant information across channels, we propose IILP, a two-step pooling strategy, *Intra-lobe Pooling* followed by *Inter-lobe Concatenation*, illustrated in Figure 4. Given an encoder block output $\mathbf{Z}^{enc,l} \in \mathbb{R}^{T \times D \times d_{model}}$, where $T$, $D$, and $d_{model}$ denote the temporal length, number of EEG channels, and embedding dimension, respectively.

We first average-pool $Z^{enc}$ along the temporal axis to aggregate information from all time steps:

$$\tilde{\mathbf{V}}_d^l = \frac{1}{T}\sum_{t=1}^{T}\mathbf{Z}_{t,d}^{enc,l}, \quad d = 1, \ldots, D. \tag{11}$$

*Intra-lobe Pooling.* We partition EEG channels into $n$ functional brain lobes, such as frontal and occipital lobes, denoted as $\mathcal{P} = \{P_1, \ldots, P_n\}$. To suppress redundancy within each channel group $P$, we compute lobe-level embeddings by averaging corresponding channel embeddings:

$$V_k^l = \frac{1}{|P_k|} \sum_{d \in P_k} \widetilde{\mathbf{V}}_d^l, \quad k = 1, \ldots, n. \tag{12}$$

*Inter-lobe Concatenation.* To leverage discriminative features across different brain lobes, we concatenate the lobe embeddings into a joint representation:

$$v^l = \text{concat}(V_1^l, \ldots, V_n^l), \tag{13}$$

where $v^l \in \mathbb{R}^{nd_{model}}$ indicates the aggregation vector for encoder block $l$. Repeating the above IILP process across all encoder blocks and stacking the results yields the final representation:

$$v = \text{concat}(v^1, \ldots, v^L), \tag{14}$$

where $L$ is the number of encoder blocks. Finally, we use a multilayer perceptron as the classifier on the representation obtained at different granularities to predict the task-specific class from the set $\mathcal{C}$.

## 3 Experiments

### 3.1 Pre-training

**Datasets** NEURIPT is pre-trained using more than 2,000 hours of data collected from public datasets, with the eight downstream datasets explicitly excluded. For more details on pre-training datasets, please refer to Appendix E.1.

**Preprocessing** To ensure a fair comparison, we follow the data pre-processing pipeline in CBraMod [21]. EEG recordings with a total duration less than 5 minutes were removed, and we discarded the first and last minute of each remaining session to mitigate boundary artifacts. Signals were re-referenced using 20 bipolar channels in the canonical "double banana" montage (e.g., FP1–F7, F7–T3, ..., P4–O2). For further details on preprocessing, please refer to Appendix E.1.2.

**Settings** We implemented NEURIPT using Python 3.9.19 and PyTorch 2.3.0 with CUDA 12.1 and cuDNN 8902. Pre-training stage was trained using the AdamW optimizer combined with the OneCycle learning rate strategy [29] (upper learning rate 3e-4, divided factor 25, final divided factor 1e4, and cosine annealing strategy). The pre-training process was conducted for approximately 400K steps, employing an effective batch size of 480 and bfloat16 mixed-precision training on eight NVIDIA GeForce RTX 4090 GPUs. For more details on implementation settings and hyperparameters, please refer to Appendix D and Table 13.

Table 1: Overview of downstream BCI tasks and datasets.

| BCI Tasks | Datasets | Rate | Channels Used | Duration | Samples | Label |
|---|---|---|---|---|---|---|
| I. Mental Stress Detection | MentalArithmetic | 500Hz | 20 | 5s | 1,707 | 2-class |
| II. Mental Disorder Diagnosis | Mumtaz2016 | 256Hz | 20 | 5s | 6,963 | 2-class |
| III. P300 | PhysioNetP300 | 2048Hz | 20 | 2s | 21,179 | 2-class |
| IV. Sleep Staging | Sleep-EDFx | 100Hz | 2 | 30s | 457,652 | 5-class |
| V. Emotion Recognition | SEED-V | 1000Hz | 20 | 1s | 115,001 | 5-class |
| VI. Motor Imagery Task | BCIC-IV-2A | 250Hz | 16 | 4s | 5,184 | 4-class |
| VII. Abnormal Detection | TUAB | 250Hz | 20 | 10s | 409,455 | 2-class |
| VIII. Event Type Classification | TUEV | 250Hz | 20 | 5s | 112,491 | 6-class |

### 3.2 Downstream BCI Tasks

**BCI Tasks and Datasets** To comprehensively assess the effectiveness of NEURIPT, we evaluated it across eight diverse BCI datasets spanning multiple downstream task categories. A summary of all tasks and corresponding datasets is provided in Table 1 and detailed in Appendix E.2.1. To ensure fair comparison, we adopt the same data processing protocols as CBraMod [21]. Additional details on each downstream dataset and preprocessing pipeline are presented in Appendix E.2.

**Baselines and Metrics** All the baselines and metrics are detailed in Appendix D.2.

Table 2: The results of NEURIPT on various datasets. Figure 5 demonstrates the visual comparison.

| Datasets | Methods | Balanced Accuracy | Cohen's Kappa / AUC-PR | Weighted F1 / AUROC |
|---|---|---|---|---|
| MentalArithmetic | BIOT [NeurIPS23][17] | 68.75 | 60.04 | 75.36 |
| | LaBraM [ICLR24][18] | 69.09 | 59.99 | 77.21 |
| | CBraMod [ICLR25][21] | 72.56 | 62.67 | 79.05 |
| | **NEURIPT (Ours)** | **86.46 (+13.90)** | **78.27 (+15.60)** | **91.11 (+12.06)** |
| Mumtaz2016 | BIOT [NeurIPS23][17] | 93.58 | 97.36 | 97.58 |
| | LaBraM [ICLR24][18] | 94.09 | 97.98 | 97.82 |
| | CBraMod [ICLR25][21] | 95.60 | 99.23 | 99.21 |
| | **NEURIPT (Ours)** | **98.03 (+2.43)** | **99.81 (+0.58)** | **99.79 (+0.58)** |
| *PhysioP300 | BENDR [15] | 61.14 | 22.27 | 65.88 |
| | BIOT [NeurIPS23][17] | 54.85 | 9.68 | 53.08 |
| | LaBraM [ICLR24][18] | 64.77 | 29.35 | 70.68 |
| | EEGPT [NeurIPS24][20] | 65.02 | 29.99 | 71.68 |
| | **NEURIPT (Ours)** | **67.31 (+2.29)** | **34.26 (+4.27)** | **76.83 (+5.15)** |
| Sleep-EDFx | BENDR [15] | 66.55 | 66.59 | 75.07 |
| | BIOT [NeurIPS23][17] | 66.22 | 64.61 | 74.15 |
| | LaBraM [ICLR24][18] | 67.71 | 67.10 | 75.92 |
| | EEGPT [NeurIPS24][20] | 69.17 | 68.57 | 76.54 |
| | **NEURIPT (Ours)** | **70.47 (+1.30)** | **77.57 (+9.00)** | **87.39 (+10.85)** |
| SEED-V | BIOT [NeurIPS23][17] | 38.37 | 22.61 | 38.56 |
| | LaBraM [ICLR24][18] | 39.76 | 23.86 | 39.74 |
| | CBraMod [ICLR25][21] | 40.91 | 25.69 | 41.01 |
| | **NEURIPT (Ours)** | **41.04 (+0.13)** | **26.29 (+0.60)** | **41.58 (+0.57)** |
| BCIC-IV-2A | BIOT [NeurIPS23][17] | 47.48 | 29.97 | 46.07 |
| | LaBraM [ICLR24][18] | 48.69 | 31.59 | 47.58 |
| | CBraMod [ICLR25][21] | 51.38 | 35.18 | 49.84 |
| | **NEURIPT (Ours)** | **55.04 (+3.66)** | **40.04 (+4.86)** | **53.76 (+3.92)** |
| TUAB | BIOT [NeurIPS23][17] | 79.59 | 87.92 | 88.15 |
| | LaBraM [ICLR24][18] | 82.58 | 92.04 | 91.62 |
| | EEGPT [NeurIPS24][20] | 79.83 | - | 87.18 |
| | NeuroLM [ICLR25][21] | 79.69 | 72.19 | 78.84 |
| | CBraMod [ICLR25][21] | 82.89 | **92.58** | **92.27** |
| | **NEURIPT (Ours)** | **82.93 (+0.04)** | 90.40 (-2.18) | 89.49 (-2.78) |
| TUEV | BIOT [NeurIPS23][17] | 52.81 | 52.73 | 74.92 |
| | LaBraM [ICLR24][18] | 66.16 | 67.45 | 83.29 |
| | EEGPT [NeurIPS24][20] | 62.32 | 63.51 | 81.87 |
| | NeuroLM [ICLR25][21] | 46.79 | 45.70 | 73.59 |
| | CBraMod [ICLR25][21] | 66.71 | 67.72 | 83.42 |
| | **NEURIPT (Ours)** | **67.61 (+0.90)** | **69.70 (+1.98)** | **84.28 (+0.86)** |

## 3.3 Main Results

We present a comparative analysis of our method against baselines across eight downstream datasets in Table 2, with the best results highlighted in bold. Moreover, we report the performance difference between our method and the best-performing baseline. Compared to the baselines, our method generally outperforms them on the majority of datasets, with the exception of Cohen's Kappa and AUROC metrics on TUAB. Notably, our method achieves significant improvements on the Mental Arithmetic, PhysioP300, and BCIC-IV-2A datasets. While the performance gains are smaller on other datasets, they remain consistent, demonstrating the robustness and general applicability of our approach. Extensive empirical evaluations on eight benchmark BCI datasets consistently demonstrate strong generalization and broad applicability across diverse EEG-based tasks. Appendix E.2 gives the specific experimental results on each task, including experimental configuration and results analysis.

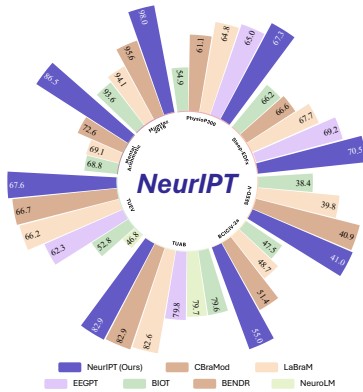

Figure 5: Models performance on various BCI downstream tasks.

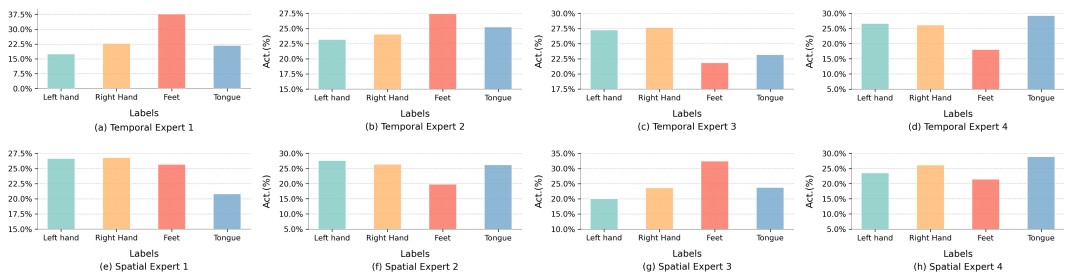

Figure 6: Analysis of expert participation (temporal and spatial) when EEG data from different classes of the BCIC-IV-2A dataset is input to the model.

## 3.4 Additional Results and Analysis

**Analysis of Spatial Relationships between EEG Channels**    To analyze spatial relationships, we presented in Figure 4 the attention scores from a spatial attention head, along with an analysis of the Pearson correlation between class logits and channel perturbations using Gaussian multiplicative noise. The attention score visualization reveals both inter- and intra-lobe interactions, which aligns with the strengths of IILP. From the perturbation analysis, we observe contralateral activation patterns in channels C3 and C4 for hand-related tasks, and a more symmetrical pattern for foot and tongue movements, in line with findings from the existing study [30]. Refer to Figure 8 and 16 in Appendix G for more visualization about the attention score and Pearson correlation in various datasets.

**Analysis of Progressive Mixture-of-Experts (PMoE)**    We analyzed how temporal and spatial experts in the PMoE architecture contribute to the prediction of the four classes in the BCIC-IV-2A dataset, with the statistics presented in Figure 6. It is observed that different classes engage varying numbers of experts, with some classes receiving attention from a larger number of experts. Refer to Figure 9-15 in Appendix G for more expert contributions across different datasets.

Table 3: Different MoE strategies across various datasets.

| MoE Strategies | No. of Experts | TUEV | MentalArithmetic | Mumtaz2016 | SEED-V | PhysioP300 | BCIC-2A |
|---|---|---|---|---|---|---|---|
| w/o Expert | [0, 0, 0, 0, 0, 0] | 65.83 | 72.92 | 93.41 | 39.14 | 64.53 | 44.44 |
| Uniform | [4, 4, 4, 4, 4, 4] | 65.91 | 70.49 | 95.00 | 39.33 | 65.66 | 41.93 |
| Shrinking | [6, 4, 4, 4, 0, 0] | 65.80 | 73.96 | 93.08 | 39.21 | 65.99 | **44.62** |
| **Progressive (Ours)** | [0, 0, 2, 4, 4, 6] | **68.94** | **75.69** | **97.07** | **39.34** | **66.58** | 44.01 |

Table 4: Different PMoE configurations across various datasets.

| PMoE Configurations | TUEV | MentalArithmetic | Mumtaz2016 | SEED-V | PhysioP300 | BCIC-2A |
|---|---|---|---|---|---|---|
| [0,0,2,4,4,6] | **68.94** | 75.69 | 97.07 | 39.34 | 66.58 | **44.01** |
| [0,0,2,3,4,5] | 67.87 | **76.39** | 96.81 | **39.88** | 66.34 | 41.58 |
| [0,0,2,4,6,8] | 66.41 | 75.69 | 96.57 | 39.11 | **67.70** | 41.15 |
| [0,0,3,6,9,12] | 66.08 | 74.83 | **97.15** | 39.35 | 66.63 | 38.19 |

**Ablation of MoE Strategies and PMoE Configurations**    Tables 3 and 4 show the ablation results of MoE strategies and PMoE configurations, respectively. When varying the number of experts across layers, our PMoE, which increases the number of experts with depth, consistently outperforms both the non-MoE baseline, the uniform-expert variant, and the shrinking MoE variant, where the number of experts decreases with depth. The alternative configs in Table 4 also yield competitive performance, which suggests that the effectiveness of the PMoE approach stems from its inherent progressive strategy, rather than relying on any specific expert allocation, demonstrating robustness across diverse progressive configurations.

Table 5: Different pooling strategies across various datasets.

| Pooling Strategies | TUEV | MentalArithmetic | Mumtaz2016 | SEED-V | PhysioP300 | BCIC-2A |
|---|---|---|---|---|---|---|
| w/o Pooling | 62.33 | 75.69 | 78.21 | 38.90 | **67.82** | 45.14 |
| Mean Pooling | 64.74 | 79.51 | 96.22 | 37.62 | 66.72 | 37.24 |
| Hemispheres | 64.45 | 81.94 | 97.82 | 39.22 | 67.11 | 43.49 |
| Coronal | 68.77 | 73.26 | 96.99 | 39.35 | 66.98 | 43.75 |
| Sagittal | 67.21 | 80.21 | 91.41 | 39.42 | 65.66 | 45.31 |
| **IILP (Ours)** | **68.94** | **86.46** | **98.03** | **41.04** | 67.31 | **55.04** |

Table 6: Ablation study on each individual component in NEURIPT. Results are based on models trained from scratch, without the time-intensive pre-training stage.

| 3D PE | PMoE | IILP | TUEV | MentalArithmetic | Mumtaz2016 | SEED-V | BCIC-IV-2A |
|-------|------|------|------|------------------|------------|--------|------------|
| ✗ | ✗ | ✗ | 51.80 | 73.36 | 91.83 | 37.82 | 32.64 |
| ✓ | ✗ | ✗ | 59.64 | 73.61 | 86.07 | 38.54 | 40.19 |
| ✗ | ✓ | ✗ | 52.79 | 74.65 | 85.58 | 37.82 | 33.59 |
| ✗ | ✗ | ✓ | 59.10 | 73.96 | 91.55 | 35.66 | 37.15 |
| ✓ | ✓ | ✗ | 62.33 | **75.69** | 78.21 | 38.90 | **45.14** |
| ✓ | ✗ | ✓ | 65.83 | 72.92 | 93.41 | 39.14 | 44.44 |
| ✗ | ✓ | ✓ | 67.72 | 74.65 | 96.56 | 35.03 | 37.59 |
| ✓ | ✓ | ✓ | **68.94** | **75.69** | **97.07** | **39.34** | 44.01 |

**Ablation of Pooling Strategies**    We compared our Intra-Inter Lobe Pooling (IILP) with no pooling as well as pooling strategies based on coronal, sagittal, and hemispheric brain region groupings. The results are presented in Table 5, demonstrating the consistent effectiveness and superiority of IILP.

**Ablation of Each Individual Component**    The ablation in Table 6 demonstrates that performance across datasets generally improves with the inclusion of each component, achieving peak accuracy when all components are combined. Notably, the IILP module significantly enhances results on the TUEV and Mumtaz2016 datasets. This improvement is presumably because epilepsy and depression classification tasks require capturing distinctive regional signal variations across different brain areas.

For BCIC-IV-2A, which primarily involves motor imagery tasks and was recorded using EEG channels located centrally with limited coverage of other brain lobes, we observed a performance decline when the 3D PE component was excluded. This suggests that motor imagery tasks are particularly sensitive to spatial information, as further supported by our channel perturbation analysis in Figure 4. Given the low-data scenario (BCIC-IV-2A includes only nine subjects), explicitly encoding physical spatial relationships becomes especially important. Moreover, excluding only IILP may degrade performance due to incomplete brain lobe coverage, limiting the model's utilization of lobe-specific information, yet incorporating it during pretraining enhances generalization (see Table 5).

Table 7: Different activation functions for EEG FMs (trained from scratch, same as Table 6).

| Activation Function | TUEV | MentalArithmetic | Mumtaz2016 | SEED-V | PhysioP300 | BCIC-2A |
|---------------------|------|------------------|------------|--------|------------|---------|
| ReLU | 64.01 | 73.96 | 96.98 | 38.37 | 66.29 | **47.31** |
| GELU | 64.02 | 73.61 | **97.14** | 38.69 | 64.38 | 46.35 |
| SwiGLU | **68.94** | **75.69** | 97.07 | **39.34** | **66.58** | 44.01 |

**Exploration of Activation Function for EEG Foundation Models (FMs)**    Different datasets exhibited distinct preferences (Table 7): ReLU performed particularly well on BCIC-2A, whereas GELU achieved competitive results on Mumtaz2016. Nonetheless, SwiGLU demonstrated the most consistent performance across multiple downstream tasks, indicating its robustness in FMs for EEG data.

**Further Results and Analysis**    For further results and analysis, please refer to Appendix B.

## 4    Conclusion

This study introduces NEURIPT, a foundation model established for diverse EEG-based neural interfaces, overcoming the challenges in learning generalizable spatio-temporal representations of EEG signals from diverse sources. Temporally, we propose Amplitude-Aware Masked Pretraining (AAMP), which selects masked segments based on signal amplitude rather than random intervals, encouraging the model to capture robust features across varying signal intensities and avoiding reliance on local interpolation. This is complemented by a Progressive Mixture-of-Experts (PMoE) architecture, which progressively incorporates specialized expert subnetworks at deeper layers to better adapt to the temporal variability inherent in EEG data. Spatially, NEURIPT utilizes the 3D physical coordinates of electrodes to support transferable embeddings across different EEG configurations and introduces Intra-Inter Lobe Pooling (IILP) during fine-tuning to effectively leverage region-specific brain activity. Comprehensive evaluations on eight benchmark EEG datasets demonstrate NEURIPT 's consistent state-of-the-art performance and strong generalization. These findings highlight NEURIPT's potential as a scalable foundation model for EEG, moving toward universal neural decoding systems.

## Acknowledgments

This work was supported in part by the Ministry of Higher Education Malaysia through the Fundamental Research Grant Scheme (FRGS/1/2023/ICT02/XMU/02/1); National Natural Science Foundation of China (62476070, 62376233); Shenzhen Science and Technology Program (JCYJ20241202123503005, GXWD20231128103232001, ZDSYS20230626091203008, KQTD20240729102154066); Department of Science and Technology of Guangdong (2024A1515011540); Natural Science Foundation of Fujian Province (2024J09001); Collaborative Innovation Platform Project for Fuzhou-Xiamen-Quanzhou National Independent Innovation Demonstration Zone (2022-P-028); Xiaomi Young Talents Program; Xiamen University Malaysia Research Fund (XMUMRF/2024-C13/IECE/0049, XMUMRF/2024-C14/IECE/0055).

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

# A  Related Work

Existing literature on EEG-based neural decoding can be broadly categorized according to the evolution from EEG setting-specific models to generalizable foundation models, as outlined below:

## A.1  Decoding EEG Signals with Task- and Setup-Specific Models

EEG decoding has evolved from classical BCI pipelines, that develop computational approaches based on the well-known EEG signatures (e.g., P300, event-related (de)synchronization, Steady-State Visual Evoked Potential (SSVEP), etc [31]) to feature engineering and statistical classifiers such as, linear discriminant analysis (LDA), or support vector (SVM), random forest, common spatial pattern (CSP) [5, 32], to more sophisticated machine learning and deep learning models tailored for specific tasks and recording setups. These models are often trained on specific datasets and optimized for particular paradigms, such as motor imagery [32], gait recognition [5], emotion recognition [33], and seizure detection [7], under fixed channel layouts, devices, or recording conditions, building upon prior knowledge of EEG characteristics. Deep learning approaches such as CNN [5], RNN [6], GNN [7], Transformers [8], and their hybrids have been widely applied to model EEG's complex spatiotemporal dynamics using spectrogram, time-frequency representation, and brain connectivity [34].

Recent work has shifted end-to-end learning paradigms that learn representations directly on raw EEG signals [35, 36, 37], reducing the dependence on handcrafted preprocessing. However, these methods still struggle with generalization across subjects, devices, or experimental conditions and require retraining when used in different settings.

## A.2  Unified EEG Signal Decoding with Pretrained Foundation Models

Motivated by the success of foundation models in natural language processing (NLP) [9, 10], recent trends in EEG research focus on developing large-scale, pretrained neural architectures that scale up and generalize across diverse EEG tasks, conditions, experimental setups, and subjects. These models are typically pre-trained using self-supervised learning on unlabeled EEG data from varied sources, enabling the extraction of task and subject-invariant representations that can be fine-tuned for a wide range of downstream EEG applications. Broadly, three main lines of approaches have emerged: (i) modeling EEG as a multivariate time series using transformer-based architectures adapted from the speech or time-series domains, such as BENDR [15] and EEG2Vec [16]; (ii) tokenizing continuous EEG signals into discrete tokens to enable large-scale pretraining analogous to language modeling, as in LaBraM [18] and NeuroLM [19]; and (iii) preserving EEG's spatiotemporal structure using specialized architectures that consider both between EEG channels and temporal dynamics from diverse datasets through neural architectural design, exemplified by recent EEG-based FMs, BIOT [17], EEGPT [20], and CBraMod [21]. These FMs reduce the need for task-specific calibration and enhance robustness and scalability in real-world BCI and EEG-based applications. CHARM [38] addressed inconsistent input channels through a learnable channel-level masking approach, and COLA [39] employed contrastive learning rather than masking-based self-supervised pretraining on large-scale audio data. These related works offer valuable complementary perspectives on enhancing EEG modeling.

While these pioneering FMs showed early promise, a few important EEG characteristics are not considered in the current FMs, such as the physical three-dimensional arrangement of electrodes, diverse EEG patterns and intensity across diverse datasets, and brain regional features are not fully taken into account. Moreover, recent successes of Mixture-of-Experts (MoE) [40, 41] based foundation models have demonstrated impressive scalability and generalization in fields like LLMs, by designing a larger network (for more parameters) with sub-networks that are only activated by relevant task (for less computation); however, their application and potential benefits remain largely unexplored in EEG decoding. This work attempts to fill the research gap for developing a robust EEG-based FM.

# B  Additional Results and Analysis

This section continues from Section 3.4 of the main text and provides further results and analysis.

Table 8: Comparison between different masking strategies (masking ratio: 50%).

| Masking Strategy | Bal. Acc. |
|---|---|
| BERT-style random masking | 0.6320 |
| Amplitude-aware masking | **0.6845** |

**Ablation of Masking Strategies**    We compare and analyze our proposed amplitude-aware masking pretraining (AAMP) strategy with BERT-style random masking in Table 8. We directly evaluate the pretrained representations by performing a downstream classification task using Support Vector Machine (SVM)[2], aiming to assess the quality of the learned representations obtained during the pretraining stage.[3] Our amplitude-based masking improves 5.25% compared to random masking, proving the effectiveness of masking based on amplitude. Figure 7 illustrates the training loss between to strategies. After 1000 steps, random masking has lower reconstruction loss, demonstrating that random masking is better for learning reconstruction. This implies that the model is better at interpolation under random masking, but ultimately performs poorly in downstream classification tasks. Refer to Figure 18, 19 and 20 of Appendix G for more visualization comparisons.

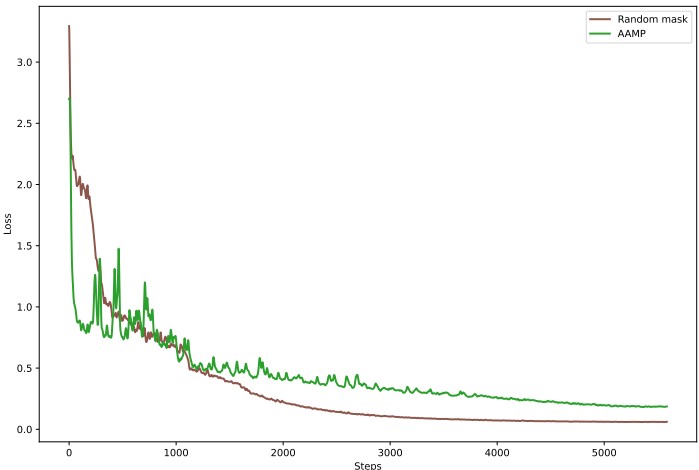

Figure 7:  Training loss between different masking strategies.

**Analysis of Masking Ratio**    We then also tested the performance of our AAMP at different masking ratios. As Table 9 shows, SVM works best at a masking ratio of 50%. Figure 17 of Appendix G shows the loss curves for different masking ratios.

Table 9: Performance of NEURIPT with different masking ratios.

| Masking Ratio (%) | Bal. Acc. |
|---|---|
| 30 | 0.5523 |
| 40 | 0.6321 |
| 50 | **0.6845** |
| 60 | 0.5596 |
| 70 | 0.5254 |

---

[2]For the classification task in Appendix B, we selected the TUEV due to its complexity and variety, consisting of six distinct classes. To manage the computational complexity of SVM, which scales quadratically with the number of data points, we randomly sampled 5000 data points from the training set and 500 data points from the test set, using a fixed random seed (520) to ensure reproducibility. Hyperparameters were optimized via grid search with 5-fold cross-validation on the sampled training data, and the best-performing combination was subsequently evaluated on the test set. Detailed ranges of hyperparameters explored during the grid search are presented in Appendix D.1 and Table 14.

[3]We use smaller models for quick tests. Specifically, we set $d_{model}$ to 256, expert hidden $\text{FFN}_{expert}$ to 128, shared expert hidden $\text{FFN}_{shared}$ to 256. The rest of the parameters remain the same as the original pre-trained model.

Table 10: Comparison between different masking strategies across diverse downstream tasks.

| Datasets | Masking Strategies | Bal. Acc. | Cohen's Kappa | Weighted F1 |
|---|---|---|---|---|
| TUEV | Random Masking | 67.35 | 67.28 | 83.37 |
| | **AAMP (Ours)** | **67.61** | **69.70** | **84.28** |
| MentalArithmetic | Random Masking | 84.38 | 77.89 | 90.98 |
| | **AAMP (Ours)** | **86.46** | **78.27** | **91.11** |
| Mumtaz2016 | Random Masking | 97.14 | **99.84** | **99.84** |
| | **AAMP (Ours)** | **98.03** | 99.81 | 99.79 |
| SEED-V | Random Masking | 33.89 | 17.07 | 33.82 |
| | **AAMP (Ours)** | **41.04** | **26.29** | **41.58** |
| PhysioP300 | Random Masking | 65.64 | 28.99 | 72.75 |
| | **AAMP (Ours)** | **67.31** | **34.26** | **76.83** |
| Sleep-EDFx | Random Masking | 69.68 | 76.82 | 87.11 |
| | **AAMP (Ours)** | **70.47** | **77.57** | **87.39** |

**Further Ablation of Masking Strategies**  We analyzed the classification performance in the preceding using SVM classifiers trained directly on representations generated under different masking strategies. Table 10 compares the downstream task performances between BERT-like random masking and our proposed AAMP, further demonstrating the advantages of our method.

Table 11: Different positional encoding strategies across various tasks. Alternative encoding strategies are included: trigonometric functions and 1D learnable embeddings employed by vanilla Transformer [9], and 2D learnable embeddings introduced in LaBraM [18].

| Datasets | Encoding Strategies | Bal. Acc. | Cohen's Kappa | Weighted F1 |
|---|---|---|---|---|
| TUEV | Trigonometric Functions | 67.72 | 68.85 | 84.03 |
| | Learnable 1D | 64.78 | 67.92 | 83.13 |
| | Learnable 2D | 63.81 | 66.63 | 82.57 |
| | **3D PE (Ours)** | **68.94** | **71.55** | **85.17** |
| MentalArithmetic | Trigonometric Functions | 74.65 | 71.84 | 83.80 |
| | Learnable 1D | 71.53 | 74.23 | 83.17 |
| | Learnable 2D | 71.52 | 64.81 | 80.83 |
| | **3D PE (Ours)** | **75.69** | **74.37** | **86.83** |
| Mumtaz2016 | Trigonometric Functions | 96.56 | 99.59 | 99.61 |
| | Learnable 1D | 96.63 | 99.21 | 99.23 |
| | Learnable 2D | 95.56 | 99.55 | 99.54 |
| | **3D PE (Ours)** | **97.07** | **99.83** | **99.84** |
| SEED-V | Trigonometric Functions | 35.03 | 18.58 | 35.08 |
| | Learnable 1D | 35.52 | 18.81 | 34.34 |
| | Learnable 2D | 37.40 | 21.54 | 37.34 |
| | **3D PE (Ours)** | **39.34** | **23.93** | **39.67** |

**Ablation of Positional Encoding Strategies**  Table 11 presents ablation studies on different position encoding methods. Our proposed 3D PE consistently outperforms both the trigonometric function embedding and the learnable 1D and 2D embeddings, underscoring its effectiveness in explicitly leveraging physical electrode positions inherent in EEG data.

Table 12: Low resource scenario. Metrics are shown as percentages relative to the full-data baseline.

| Datasets | Data Used | Bal. Acc. | Cohen's Kappa | Weighted F1 |
|---|---|---|---|---|
| TUEV | 1% | 64.16 | 72.44 | 88.08 |
| | 5% | 75.39 | 81.78 | 91.67 |
| | 10% | 80.30 | 88.64 | 95.75 |
| | 100% | 100.00 | 100.00 | 100.00 |
| Sleep-EDFx | 1% | 80.97 | 83.86 | 91.33 |
| | 5% | 86.01 | 79.82 | 90.41 |
| | 10% | 92.51 | 92.74 | 96.13 |
| | 100% | 100.00 | 100.00 | 100.00 |

**Low Data Scenario**   The performance of NEURIPT under low-resource scenarios is presented in Table 12. Our approach achieved notable classification accuracy, maintaining performance in the range of $80\% - 90\%$ with merely $10\%$ of the original data, and even with an extremely limited $1\%$ of data, performance remains competitive at $64\% - 80\%$. These results further confirm the robustness of our pretraining strategy and underscore the model's potential in few-shot and low-resource settings.

For the ablation studies in Table 6 and 7 in Section 3.4, due to time and resource constraints, the results presented are based on models trained from scratch, without the time-intensive pre-training stage.

# C Additional Methodological Details

## C.1 Backbone Architecture

We adopt Crossformer [26] as our backbone architecture, which leverages the TSA (Two-stage attention, including cross-time attention and cross-dimension attention) module to hierarchically capture alternating temporal and spatial dependencies. Additionally, the hierarchical structure of TSA, especially at deeper layers with coarser granularity, significantly reduces sequence lengths and computational demands, enabling efficient resource utilization during large-scale pre-training. Furthermore, we replace the Post-LN with Pre-LN [42] and switch the activation function from GELU [43] to SwiGLU [44] for a more stable training process, aligning with mainstream large language models such as the Llama [45, 46], Qwen [41, 47, 48], and Deepseek [40, 49] series. The mathematical formulation of our modified backbone is as follows:

**Temporal Stage** After embedding, $\mathbf{S}^{enc} \in \mathbb{R}^{T \times D \times d_{model}}$ is the encoder block $\mathbb{ENC}$'s input, where $T$ and $D$ are the number of time steps and electrode channels, respectively. Consistent with Section 2.2, $\mathbf{Z}^{l-1}$ is the output of the layer $l-1$ and $\bar{\mathbf{Z}}^l$ is the input of the layer $l$ after merging. Thus at the first layer, $\mathbf{Z}^0 = \mathbf{S}^{enc}$. For convenience, in the following, we use $\mathbf{Z}_{i,:}$ to denote the vectors of all dimensions at time step $i$, $\mathbf{Z}_{:,d}$ for those of all time steps in dimension $d$. In the temporal stage, we directly apply multi-head self-attention MSA to each dimension:

$$\check{\mathbf{Z}}^{time,l} = \text{LayerNorm}\left(\bar{\mathbf{Z}}^{l-1}\right) \tag{15}$$

$$\hat{\mathbf{Z}}^{time,l}_{:,d} = \text{LayerNorm}\left(\bar{\mathbf{Z}}^{l-1} + \text{MSA}\left(\check{\mathbf{Z}}^{time,l}_{:,d}, \check{\mathbf{Z}}^{time,l}_{:,d}, \check{\mathbf{Z}}^{time,l}_{:,d}\right)\right) \tag{16}$$

$$\mathbf{Z}^{time,l} = \text{PMoE}_1^{(l)}\left(\hat{\mathbf{Z}}^{time,l}_{:,d}\right) + \tilde{\mathbf{Z}}^{time,l}_{:,d} \tag{17}$$

where $1 \leq d \leq D$ and LayerNorm denotes pre-layer normalization [42], $\text{MSA}(\mathbf{Q}, \mathbf{K}, \mathbf{V})$ denotes the multi-head self-attention layer [9] where $\mathbf{Q}, \mathbf{K}, \mathbf{V}$ serve as queries, keys and values, PMoE denotes the Progressive Mixture-of-Experts introduced in Section 2.2, and $\tilde{\mathbf{Z}}^{time,l}_{:,d}$ denotes the residual connection result from Equation 16 but without LayerNorm. All dimensions $(1 \leq d \leq D)$ share the same MSA layer. $\mathbf{Z}^{time,l}$ denotes the output of the cross-time stage.

**Spatial Stage** Similarly to the temporal stage, we also apply the above architecture to each time step:

$$\check{\mathbf{Z}}^{dim,l} = \text{LayerNorm}\left(\mathbf{Z}^{time,l}\right) \tag{18}$$

$$\hat{\mathbf{Z}}^{dim,l}_{i,:} = \text{LayerNorm}\left(\mathbf{Z}^{time,l} + \text{MSA}\left(\check{\mathbf{Z}}^{dim,l}_{i,:}, \check{\mathbf{Z}}^{dim,l}_{i,:}, \check{\mathbf{Z}}^{dim,l}_{i,:}\right)\right) \tag{19}$$

$$\mathbf{Z}^l = \text{PMoE}_2^{(l)}\left(\hat{\mathbf{Z}}^{dim,l}_{:,d}\right) + \tilde{\mathbf{Z}}^{dim,l}_{:,d} \tag{20}$$

where $1 \leq i \leq T$ and $\tilde{\mathbf{Z}}^{dim,l}_{:,d}$ denotes the residual connection result from Equation 19 but without LayerNorm. Finally, $\mathbf{Z}^l$ is the output of the entire TSA layer $l$. Notably, in practice, at the first layer, we split Equation 2 into the following two parts to embed the temporal and spatial information separately:

$$\check{\mathbf{Z}}^{time,1} = \text{LayerNorm}\left(\mathbf{Z}^0 + \mathbf{PE}^{(t)}\right) \tag{21}$$

$$\check{\mathbf{Z}}^{dim,1} = \text{LayerNorm}\left(\mathbf{Z}^{time,1} + \mathbf{PE}^{(s)}\right) \tag{22}$$

Equations 21, 22 correspond to Equations 15, 18, respectively, and represent special cases at the first layer. Here, multi-head self-attention MSA is implemented through FlashAttention-2 by [50] to achieve faster speeds.

**Hierarchical Attention Modules** In each encoder layer (except the first layer), every adjacent $m_l$ vectors are merged to produce representations at progressively coarser temporal scales. Then, a TSA layer captures dependencies at this new scale. Formally, the encoder process $\mathbb{ENC}$ for layer $l$ is defined as:

$$\begin{cases} l = 1: & \bar{\mathbf{Z}}^{enc,0} = \mathbf{Z}^{enc,0}, \\ l > 1: & \bar{\mathbf{Z}}^{enc,l}_{i,d} = \mathbf{M}\left[\mathbf{Z}^{enc,l}_{(i-1)m_l+1,d} \cdot \mathbf{Z}^{enc,l}_{(i-1)m_l+2,d} \cdots \mathbf{Z}^{enc,l}_{im_l,d}\right], \end{cases} \tag{23}$$

where $1 \leq i \leq \left\lceil \frac{T_{l-1}}{m_l} \right\rceil$, $1 \leq d \leq D$, $\mathbf{Z}^{enc,l}$ denotes the output of the $l$-th encoder layer, $\mathbf{M} \in \mathbb{R}^{d_{model} \times m_l d_{model}}$ is a learnable matrix for dynamic segment merging, "$\cdot$" denotes the concatenation operation, $T_{l-1}$ denotes the number of segments in layer $l-1$, and if not divisible by $m_l$, padding is applied to $\mathbf{Z}^{enc,l-1}$ accordingly.

## C.2 Additional Masking Strategy

Besides Amplitude-Aware Masking Pretraining (AAMP), we incorporate an additional basic masking strategy proposed in BERT [22]: $80\%$ of the selected tokens are replaced with the `[mask]` token, $10\%$ are replaced with a randomly sampled embedding from a predefined set $\mathbf{S}$ to mitigate overfitting, and the remaining $10\%$ are left unchanged to prevent the model from overly relying on the `[mask]` token.

## C.3 Mixture-of-Experts

We incorporate an auxiliary loss consisting of an importance loss $\mathcal{L}^{(l)}_{\text{importance}}$ and a load-balancing loss $\mathcal{L}^{(l)}_{\text{balance}}$ of layer $l$ proposed by Google Brain [27]. This auxiliary loss promotes equitable expert allocation, improving model efficiency and generalization:

$$\mathcal{L}_{\text{aux}} = \sum_{l=1}^{L} \mathcal{L}^{(l)}_{\text{importance}} + \mathcal{L}^{(l)}_{\text{balance}}. \tag{24}$$

## C.4 Balanced Binary Cross Entropy

We applied Bal-BCE [51], a logit-adjusted balanced binary cross-entropy loss that has achieved promising results in long-tailed image datasets. Here, we explore similar approaches for imbalanced BCI downstream datasets. Let $\pi_{\mathbf{y}_i} = n_{\mathbf{y}_i}/N$ represent the class distribution for $\mathbf{y}_i$, where the bias term of the logit $\mathbf{z}_{\mathbf{y}_i}$ is given by $\mathcal{B}^{\text{bce}}_{\mathbf{y}_i} = \log \pi_{\mathbf{y}_i} - \log(1 - \pi_{\mathbf{y}_i})$. Based on this logit bias adjustment, the loss function we adopt during the fine-tuning stage is as follows:

$$\mathcal{L}_{\text{Bal-BCE}} = -\sum_{\mathbf{y}_i \in \mathcal{C}} w_i \left[ \mathbb{1}(\mathbf{y}_i) \cdot \log \sigma \left( \mathbf{z}_{\mathbf{y}_i} + \mathcal{B}^{\text{bce}}_{\mathbf{y}_i} \right) + (1 - \mathbb{1}(\mathbf{y}_i)) \cdot \log \left( 1 - \sigma \left( \mathbf{z}_{\mathbf{y}_i} + \mathcal{B}^{\text{bce}}_{\mathbf{y}_i} \right) \right) \right], \tag{25}$$

where $\mathbb{1}$ is 1 if the condition is true and $\sigma(x) = 1/(1 + e^{-x})$ indicates the sigmoid operation.

# D  Implementation Details

## D.1  Hyperparameters and Settings

Table 13: Hyperparameters for NEURIPT pre-training.

| | Hyperparameters | Settings |
|---|---|---|
| EEG sample | Channels | 20 |
| | Data length | 256 |
| | Dynamic masking ratio | [20,35,50] |
| Model Architecture | Input dimension | 768 |
| | Output dimension | 768 |
| | Feed-forward dimension | 768 |
| | Heads | 8 |
| | Merge layers | [1,4,1,2,1,2] |
| PMoE Configuration | Hidden dimension of expert | 512 |
| | Shared expert hidden dimension | 768 |
| | Encoder expert | [0,2,2,4,4,6] |
| | Decoder expert | [0,0,0,0,0,0] |
| | Cross expert | [0,0,0,0,0,0] |
| | Top-$k\%$ | 0.5 |
| | Noise std | 0.001 |
| | W importance | 0.008 |
| | Auxiliary loss weight | 0.8 |
| Pre-training | Epochs | 40 |
| | Batch size | 60 |
| | Dropout | 0.1 |
| | Optimizer | AdamW |
| | Maximum learning rate | 3e-4 |
| | Div factor | 25 |
| | Final div factor | 1e4 |
| | Warm up ratio | 0.15 |
| | AdamW $(\beta_1, \beta_2)$ | (0.9, 0.98) |
| | Weight decay | 5e-3 |
| | Scheduler | CosineAnnealingLR |
| | Cosine cycle epochs | 40 |
| | Minimal learning rate | 1e-5 |
| | Clipping gradient | 100 |
| | Weights init | Kaiming normalization |
| | Seed | 520 |

Table 14: SVM grid search hyperparameters (ablation study only).

| Hyperparameters | Settings |
|---|---|
| kernel | ["linear", "rbf"] |
| C | [0.1, 1, 10, 100] |
| gamma (only rbf) | ["scale", "auto", 0.001, 0.005, 0.01, 0.05, 0.1, 0.5] |
| tol | [1e-4, 1e-3] |
| class weight | [None, "balanced"] |

Hyperparameters of pre-training are reported in Table 13, while Table 14 presents the parameters we used in SVM grid search discussed in Appendix B. For complete details of the settings used on downstream datasets, please refer to Appendix E.2. During fine-tuning, we conducted a grid search over the hyperparameters within the ranges specified in Table 15. Owing to the inherent variability and heterogeneity of EEG datasets, we observed that, for certain datasets, reinitializing model weights was essential for NEURIPT to effectively adapt and fine-tune to downstream tasks (i.e., SEED-V, and BCIC-IV-2A). This behavior is likely attributed to a representational mismatch between the pretrained model's and the target task. To address this issue, we performed reinitialization when validation performance showed early saturation during fine-tuning. By resetting the model parameters, this enables the network to escape suboptimal representational basins. Benefiting from distributed data parallelism and FlashAttention-2 [50], the entire pre-training procedure was completed within 30 hours. The subsequent fine-tuning on downstream tasks typically required only a few minutes to several hours, depending on the scale of the specific datasets involved.

Table 15: Hyperparameter tuning ranges used for diverse BCI tasks.

| Hyperparameters | Settings |
|---|---|
| Dropout | 0 ~ 0.5 |
| Effective batch size | [4, 8, 16, 32, 64] |
| Train Epochs | 10 ~ 150 |
| Maximum learning rate | $10^{-6} \sim 10^{-4}$ |
| Div factor | 1 ~ 5 |
| Final div factor | 1 ~ 1e5 |
| Warm up ratio | 0 ~ 0.3 |
| Class layers | 1 ~ 6 |
| Merge layers | [[1 2 1 2 1 2], [1 4 1 2 1 2], [1 1 8 1 2 1]] |
| Encoder expert | [[0 0 0 0 0 0], [0 0 2 4 4 6]] |
| Auxiliary loss weight | 0.8 ~ 1 |
| Noise std | 0.001 ~ 0.1 |
| W importance | 0.001 ~ 0.1 |
| BCE K | 0 ~ 0.3 |
| Freeze epochs | 0 ~ 50 |

Table 16: The number of parameters in different methods.

| Methods | Activated Params. |
|---|---|
| EEGNet [52] | 0.003M |
| EEGConformer [53] | 0.55M |
| SPaRCNet [54] | 0.79M |
| ContraWR [23] | 1.6M |
| CNN-Transformer [55] | 3.2M |
| FFCL [6] | 2.4M |
| ST-Transformer [8] | 3.5M |
| BIOT [NeurIPS23] [17] | 3.2M |
| LaBraM-Base [ICLR24] [18] | 5.8M |
| LaBraM-Huge [ICLR24] [18] | 369.0M |
| EEGPT [NeurIPS24] [20] | 25.0M |
| CBraMod [ICLR25] [21] | 4.2M |
| NeuroLM-XL [ICLR25] [19] | 1696.0M |
| NEURIPT (Ours) | 73.5M |

## D.2 Baselines and Metrics

**Baselines** We compare NEURIPT against eight non-foundation methods— BENDR [15], EEG-Net [52], EEGConformer [53], SPaRCNet [54], ContraWR [23], CNN-Transformer [55], FFCL [6], and ST-Transformer [8] —and five foundational model baselines: BIOT [17], LaBraM [18], EEGPT [20], NeuroLM [19], and CBraMod [21]. For any model without published downstream results, we directly followed the results reported in CBraMod [21] and EEGPT [20]. Strictly following the settings of CBraMod, we fine-tune BIOT and LaBraM based on their open-source code and pre-trained weights, unless their experimental results have already been reported in the original papers. Table 16 demonstrates the number of activated parameters for different baselines.

**Metrics** For binary classification tasks, we report Balanced Accuracy, Area Under the Precision–Recall Curve (AUC-PR), and Area Under the ROC curve (AUROC). For multi-class tasks, we use Balanced Accuracy, Cohen's Kappa, and Weighted F1 score. Note that for PhysioP300, we replace AUC-PR with Cohen's Kappa to match the evaluation protocol of our EEGPT baseline [20]. More details about the metrics we used:

- **Balanced Accuracy**: the average recall across all classes, mitigating class imbalance;
- **AUC-PR**: the area under the precision-recall curve;
- **AUROC**: the area under the receiver operating characteristic curve;
- **Cohen's Kappa**: a measure of inter-rater agreement that accounts for chance, computed based on observed versus expected agreement in a confusion matrix;
- **Weighted F1 Score**: the harmonic mean of precision and recall, weighted by class frequency.

# E    Dataset Description

## E.1    Pre-training Datasets

Table 17 presents the statistics of the datasets used for pre-training NEURIPT.

Table 17: Overview of pretraining datasets.

| Datasets | Subject | Total Time |
|---|---|---|
| Emobrain | 16 | 4.94h |
| Grasp and Lift EEG challenge | 12 | 11.72h |
| Inria BCI Challenge | 26 | 29.98h |
| EEG Motor Movement/Imagery | 109 | 47.30h |
| Raw EEG Data | 58 | 34.35h |
| Resting State EEG Date | 22 | 3.04h |
| SEED-Series | 46 | 166.75h |
| Siena Scalp EEG Database | 14 | 30.47h |
| SPIS Resting State Dataset | 10 | 0.83h |
| Target Versus Non-Target | 50 | 16.00h |
| TUAR | 213 | 92.22h |
| TUEP | 200 | 591.22h |
| TUSZ | 315 | 1,138.53h |
| TUSL | 38 | 20.59h |

### E.1.1    Description of Pre-training Datasets

**Emobrain [56]**    A multimodal emotion dataset contains fNIRS and 64-channel EEG recordings at a sampling rate of 1024 Hz. EEGs are recorded by the Biosemi Active 2 acquisition system, including 16 subjects.Emotional responses were induced using a subset of the IAPS dataset.

**Grasp and Lift EEG challenge [57]**    A dataset containing 32-channel EEG recordings at a sampling rate of 500 Hz. It includes data from 12 subjects performing grasp-and-lift (GAL) trials. EEG signals were recorded using an EEG cap in conjunction with a BrainAmp EEG signal amplifier.

**Inria BCI Challenge [58]**    A P300-Speller dataset that includes 26 subjects with 56-channel EEG recordings at a sampling rate of 600 Hz using Ag/AgCl EEG sensors (VSM-CTF compatible system).

**EEG Motor Movement/Imagery [59]**    A motor imagery dataset comprises EEG recordings from 109 subject performing 2 baseline tasks (eyes-open and eyes-closed), motor movement and motor imagery (both fists or both feet).The EEGs were collected using 64 channels at a sampling rate of 160 Hz with the BCI2000 system.

**Raw EEG Data [60]**    A dataset containing 64-channel EEG recordings sampled at 256 Hz, recorded during the reported Information-Integration categorization task and the reported multidimensional Rule-Based categorization task.

**Resting State EEG Date [61]**    An EEG dataset (64 channels, 256 Hz) containing 22 subjects for a resting task of 8 mins with 4 mins of eyes closed and 4 mins of eyes open using active Ag/AgCl electrodes either mounted in a BioSemi electrode cap or via freestanding electrodes.

**SEED-Series [62, 63, 64]**    A series of datasets, including SEED, SEED-IV, SEED-GER, SEED-FRA. All EEG signals were recorded from 62 channels at a sampling rate of 1000 Hz with the ESI NeuroScan System in response to videos.

**Siena Scalp EEG Database [65]**    A database consists of 31-channel EEG recordings from 14 patients, collected at a sampling rate of 512 Hz using EB Neuro and Natus Quantum LTM amplifiers, along with reusable silver/gold cup electrodes.

**SPIS Resting State Dataset [66]**    A dataset including recordings from 10 subjects, with 2.5-minute EEG segments collected in both eyes-closed and eyes-open resting states, prior to a 105-minute session of Sustained Attention to Response Task (SART) using fixed-sequence and varying ISIs. Monopolar EEG activity (64 channels, 2048 Hz) was recorded using 64 Ag/AgCl active electrodes.

**Target Versus Non-Target [67]**    A P300 dataset including 32-channel EEG signals at a sampling rate of 512 Hz from 50 subjects.

**TUAR [68]**    This subset of TUEG contains EEG recordings annotated with 5 different artifacts, recorded from 23 channels at a sampling rate of 256 Hz.

**TUEP [69]**    This subset of the TUEG comprising EEG recordings from 100 subjects with epilepsy and 100 subjects without epilepsy, as determined by a certified neurologist. The EEG was recorded using 19-23 channels at a sampling rate of 256 Hz.

**TUSZ [70]**    This corpus contains EEG signals that have been manually annotated for seizure events (including start time, stop time, channel, and seizure type). The EEG was recorded using 19-23 channels at a sampling rate of 256 Hz.

**TUSL [71]**    This is another subset of the TUEG containing annotations of slowing events, recorded from 23 channels at 256 Hz. This corpus has been used to study common error modalities in automated seizure detection.

### E.1.2    Preprocessing of Pre-training Datasets

In preprocessing, a 0.1–30 Hz band-pass filter was applied to suppress low- and high-frequency noise, followed by a notch filter to eliminate powerline interference. Data were resampled to 64 Hz and segmented into non-overlapping 4-second windows.

To further improve data quality, we removed samples with any signal exceeding 100 µV in absolute amplitude, or with values consistently below 3 µV or above 3 µV across all time points in any channel, as these likely reflect artifacts or flatlining. Subsequently, data were normalized using A-law companding (from digital signal processing) with $A = 0.25$ was used to adjust the dynamic range all EEG signals to ensure consistent scaling. After pre-processing, we retained 2,219,455 EEG segments, totaling over 2,100 hours of clean data for pre-training.

### E.2    Downstream Datasets

### E.2.1    Description of Downstream Datasets

**I. Mental Stress Detection**    This task focuses on identifying an individual's stress level using EEG signals. The MentalArithmetic dataset [72, 73] contains EEG recordings from 36 subjects of varying genders and ages, collected both before and during the performance of mental arithmetic tasks.

**II. Mental Disorder Diagnosis**    This task aims to categorize mental health states based on EEG activity. The Mumtaz2016 dataset [74] comprises EEG recordings from 34 patients diagnosed with major depressive disorder and 30 healthy control subjects.

**III. P300 Task**    This task involves detecting the P300 wave, an event-related potential reflecting cognitive processes such as attention, stimulus evaluation, and target recognition. The PhysioNetP300 dataset [75] provides EEG recordings commonly used for benchmarking P300-based brain–computer interface studies.

**IV. Sleep Staging Detection**    This task aims to classify sleep stages based on polysomnographic EEG recordings. The SleepEDFx dataset [76] contains 197 whole-night recordings and is widely used as a benchmark for automated sleep staging.

**V. Emotion Recognition**    This task concerns the detection and interpretation of emotional states from EEG data. The SEED-V dataset [3] provides EEG recordings collected while subjects watched emotionally evocative videos.

**VI. Motor Imagery Task**    This task focuses on decoding motor imagery activities from EEG signals. The BCI Competition IV-2A dataset [4] consists of EEG recordings from 9 healthy subjects performing four types of motor imagery tasks.

**VII. Abnormal Detection**  This task involves identifying abnormal neural patterns in EEG data. We followed CBraMod [21], TUAB [2] was chosen as the downstream dataset for this task. The TUAB dataset [2], a large-scale clinical EEG corpus, provides recordings annotated for abnormal events and has been widely used for automated abnormality detection.

**VIII. Event Type Classification**  This task focuses on categorizing EEG segments into distinct event types. Similarly, following CBraMod [21], TUEV [2] was chosen as the downstream dataset for this task. The TUEV dataset [2] includes EEG recordings annotated for epileptic and non-epileptic events, making it a standard resource for event-type classification studies.

### E.2.2  Mental Stress Detection: MentalArithmetic (2-class)

MentalArithmetic [72, 73] is an EEG dataset that contains recordings of 36 subjects of different genders and ages before and during the performance of mental arithmetic tasks. The EEGs are recorded from 20 silver/silver chloride electrodes placed according to the international 10-20 system at 500Hz sampling rate. All EEG recordings during mental arithmetic are labeled "under stress". The ones that are not during mental arithmetic are labeled "no stress". We present the analysis of expert participation on this dataset in Figure 10.The channel relationships are visualized in Figure 8.

Table 18: Comparison of different methods on mental stress detection (MentalArithmetic, 2-class).

| Method | Balanced Accuracy | AUC-PR | AUROC |
|---|---|---|---|
| EEGNet [52] | $0.6770 \pm 0.0116$ | $0.5763 \pm 0.0102$ | $0.7321 \pm 0.0108$ |
| EEGConformer [53] | $0.6805 \pm 0.0123$ | $0.5829 \pm 0.0134$ | $0.7424 \pm 0.0128$ |
| SPaRCNet [54] | $0.6879 \pm 0.0107$ | $0.5825 \pm 0.0193$ | $0.7418 \pm 0.0132$ |
| ContraWR [23] | $0.6631 \pm 0.0097$ | $0.5787 \pm 0.0164$ | $0.7332 \pm 0.0082$ |
| CNN-Transformer [55] | $0.6779 \pm 0.0268$ | $0.5777 \pm 0.0285$ | $0.7258 \pm 0.0336$ |
| FFCL [6] | $0.6798 \pm 0.0142$ | $0.5786 \pm 0.0266$ | $0.7330 \pm 0.0198$ |
| ST-Transformer [8] | $0.6631 \pm 0.0173$ | $0.5672 \pm 0.0259$ | $0.7132 \pm 0.0174$ |
| BIOT [NeurIPS23] [17] | $0.6875 \pm 0.0186$ | $0.6004 \pm 0.0195$ | $0.7536 \pm 0.0144$ |
| LaBraM [ICLR24] [18] | $0.6909 \pm 0.0125$ | $0.5999 \pm 0.0155$ | $0.7721 \pm 0.0093$ |
| CBraMod [ICLR25] [21] | $0.7256 \pm 0.0132$ | $0.6267 \pm 0.0099$ | $0.7905 \pm 0.0073$ |
| **NEURIPT (Ours)** | $\mathbf{0.8646} \pm 0.0107$ | $\mathbf{0.7827} \pm 0.0197$ | $\mathbf{0.9111} \pm 0.0163$ |

**Preprocessing**  All data were first converted to uniform units. A 50 Hz notch filter was applied to attenuate power-line noise, followed by resampling the data to 64 Hz. A global average reference was then used. And following CBraMod [21], the data were segmented into 5-seconds.

**Experimental Configuration**  Following CBraMod [21], subject 1 to 28 are set to training set, subject 29 to 32 are set to validation set and subject 33 to 36 are set to test set.

**Results**  As shown in Table 18, NEURIPT achieves the state-of-the-art performance and gets a great improvement. NEURIPT performs 13.9 % better than the best baseline in balanced accurary, 15.6 % better than the best baseline in AUC-PR and 12.06 % better than the best baseline in AUROC.

### E.2.3  Mental Disorder Diagnosis: Mumtaz2016 (2-class)

Mumtaz2016 [74] consists of EEG recordings from 34 patients diagnosed with major depressive disorder (MDD) and 30 healthy subjects (H). All EEGs recorded from 19 electrodes placed according to the international 10–20 system at 256Hz sampling rate. The dataset collects three sessions, including eyes-open session, eyes-closed session, and task session. We present the analysis of expert participation on this dataset in Figure 11. The channel relationships are visualized in Figure 8.

**Preprocessing**  The data were first converted to uniform units. A 50 Hz notch filter was then applied to attenuate power-line noise. After that, the data were resampled to 64 Hz and re-referenced using a global average reference. Following CBraMod [21], the data were segmented into 5-seconds.

**Experimental Configuration**  In our experiment, we only uses eyes-open session and eyes-closed session. Similarly, following CBraMod [21], 24 MDD and 19 H are used for training,5 MDD and 4 H are used for validation, and 5 MDD and 5 H are used for test.

Table 19: Comparison of different methods on mental disorder diagnosis (Mumtaz2016, 2-class).

| Method | Balanced Accuracy | AUC-PR | AUROC |
|---|---|---|---|
| EEGNet [52] | $0.9232 \pm 0.0104$ | $0.9626 \pm 0.0095$ | $0.9639 \pm 0.0093$ |
| EEGConformer [53] | $0.9308 \pm 0.0117$ | $0.9684 \pm 0.0105$ | $0.9702 \pm 0.0101$ |
| SPaRCNet [54] | $0.9316 \pm 0.0095$ | $0.9754 \pm 0.0065$ | $0.9781 \pm 0.0083$ |
| ContraWR [23] | $0.9195 \pm 0.0115$ | $0.9589 \pm 0.0102$ | $0.9621 \pm 0.0092$ |
| CNN-Transformer [55] | $0.9305 \pm 0.0068$ | $0.9757 \pm 0.0074$ | $0.9742 \pm 0.0059$ |
| FFCL [6] | $0.9314 \pm 0.0038$ | $0.9717 \pm 0.0021$ | $0.9753 \pm 0.0033$ |
| ST-Transformer [8] | $0.9135 \pm 0.0103$ | $0.9578 \pm 0.0086$ | $0.9594 \pm 0.0059$ |
| BIOT [NeurIPS23] [17] | $0.9358 \pm 0.0052$ | $0.9736 \pm 0.0034$ | $0.9758 \pm 0.0042$ |
| LaBraM [ICLR24] [18] | $0.9409 \pm 0.0079$ | $0.9798 \pm 0.0093$ | $0.9782 \pm 0.0057$ |
| CBraMod [ICLR25] [21] | $0.9560 \pm 0.0056$ | $0.9923 \pm 0.0032$ | $0.9921 \pm 0.0025$ |
| **NEURIPT (Ours)** | $\mathbf{0.9803} \pm 0.0062$ | $\mathbf{0.9981} \pm 0.0044$ | $\mathbf{0.9979} \pm 0.0045$ |

**Results**    As shown in Table 19, NEURIPT achieves the state-of-the-art performance. NEURIPT performs 2.43 % better than the best baseline in balanced accuracy.

### E.2.4  P300 Task: PhysioNetP300 (2-class)

PhysioNetP300 [75] is typical P300 task dataset. Each record in the dataset contains the signals, triggers and annotations corresponding to a single run. In this dataset, each subject was asked to spell a total of 20 characters using a Donchin speller. The target characters were randomly selected before the start of the run. Each row and column of a standard 6x6 character matrix was randomly augmented for 100 ms at 50 ms intervals with approximately 20 flashes. During this time, subjects need to focus on the target and count the number of times the target was highlighted. When the target was highlighted, we labeled it as "Target". Otherwise, we label it as "Non-target". We present the analysis of expert participation on this dataset in Figure 12. The channel relationships are visualized in Figure 8.

Table 20: The results of different methods on P300 task (PhysioNetP300, 2-class).

| Methods | Balanced Accuracy | Cohen's Kappa | AUROC |
|---|---|---|---|
| BENDR [15] | $0.6114 \pm 0.0118$ | $0.2227 \pm 0.0237$ | $0.6588 \pm 0.0163$ |
| BIOT [NeurIPS23] [17] | $0.5485 \pm 0.0325$ | $0.0968 \pm 0.0647$ | $0.5308 \pm 0.0333$ |
| LaBraM [ICLR24] [18] | $0.6477 \pm 0.0110$ | $0.2935 \pm 0.0227$ | $0.7068 \pm 0.0134$ |
| EEGPT [NeurIPS24] [20] | $0.6502 \pm 0.0063$ | $0.2999 \pm 0.0139$ | $0.7168 \pm 0.0051$ |
| **NEURIPT (Ours)** | $\mathbf{0.6731} \pm 0.0045$ | $\mathbf{0.3426} \pm 0.0074$ | $\mathbf{0.7683} \pm 0.0039$ |

**Preprocessing**    The data were first converted to uniform units. A 60 Hz notch filter was applied to attenuate power-line noise. The signals were then resampled to 64 Hz and re-referenced using a global average reference. Following the preprocessing steps used in EEGPT [20], the data were segmented into 2-seconds starting at 0.7 seconds before the onset of the flicker stimulus.

**Experimental Configuration**    Following EEGPT [20], subjects 8, 10, and 12 were removed and the data from the remaining subjects were retained. We split the subjects into training set and testing set randomly.

**Results**    As shown in Table 20, NEURIPT achieves the state-of-the-art performance. NEURIPT performs better than the best baseline in all 3 evaluation metrics.

### E.2.5  Sleep Stage Detection: SleepEDFx (5-class)

SleepEDFx [76] is a dataset that contains 197 (78 healthy subjects) whole-night polysomnographic sleep recordings, including EEG, EOG, chin EMG, and event markers. The sampling rate are 100Hz of the EEG and EOG channels, the ohter channels are 1Hz. We present the analysis of expert participation on this dataset in Figure 13. The channel relationships are visualized in Figure 8.

Table 21: The results of different methods on sleep stage detection (SleepEDFx, 5-class).

| Methods | Balanced Accuracy | Cohen's Kappa | Weighted F1 |
|---|---|---|---|
| BENDR [15] | $0.6655 \pm 0.0043$ | $0.6659 \pm 0.0043$ | $0.7507 \pm 0.0029$ |
| BIOT [NeurIPS23] [17] | $0.6622 \pm 0.0013$ | $0.6461 \pm 0.0017$ | $0.7415 \pm 0.0010$ |
| LaBraM [ICLR24] [18] | $0.6771 \pm 0.0022$ | $0.6710 \pm 0.0006$ | $0.7592 \pm 0.0005$ |
| EEGPT [NeurIPS24] [20] | $0.6917 \pm 0.0069$ | $0.6857 \pm 0.0019$ | $0.7654 \pm 0.0023$ |
| **NEURIPT (Ours)** | $\mathbf{0.7047} \pm 0.0041$ | $\mathbf{0.7757} \pm 0.0015$ | $\mathbf{0.8739} \pm 0.0013$ |

**Preprocessing**   The data were first converted to uniform units. They were then resampled to 64 Hz and re-referenced using a global average reference. Following EEGPT [20], the data were segmented into 30-seconds.

**Experimental Configuration**   We use the whole sleep-cassette (SC) and sleep-telemetry (ST) dataset that includes 197 subjects. Following EEGPT [20], subjects were randomly divided according to the ratio of 60.

**Results**   As shown in Table 21, NEURIPT achieves the state-of-the-art performance. NEURIPT performs better than the best baseline in all 3 evaluation metrics. In particular, NEURIPT gets a great improvement in Cohen's Kappa and Weighted F1.

### E.2.6   Emotion Recognition: SEED-V (4-class)

The SEED-V dataset [3] comprises EEG and eye movement recordings from 16 participants during emotion elicitation tasks. Each participant completed three sessions on separate days, with each session containing 15 trials—three trials for each of five emotion categories: happy, sad, neutral, disgust, and fear. EEG signals were recorded using a 62-channel ESI NeuroScan system at a sampling rate of 1000 Hz. The dataset provides both raw EEG data and extracted differential entropy (DE) features across five frequency bands, facilitating various analyses in emotion recognition research. We also present the analysis of expert participation on the this dataset in Figure 14, and report the Pearson correlation between class logits and channel perturbations induced by Gaussian multiplicative noise in Figure 16. The channel relationships are visualized in Figure 8.

Table 22: The results of different methods on emotion recognition (SEED-V, 5-class).

| Methods | Balanced Accuracy | Cohen's Kappa | Weighted F1 |
|---|---|---|---|
| EEGNet [52] | $0.2961 \pm 0.0102$ | $0.1006 \pm 0.0143$ | $0.2749 \pm 0.0098$ |
| EEGConformer [53] | $0.3537 \pm 0.0112$ | $0.1772 \pm 0.0174$ | $0.3487 \pm 0.0136$ |
| SPaRCNet [54] | $0.2949 \pm 0.0078$ | $0.1121 \pm 0.0139$ | $0.2979 \pm 0.0083$ |
| ContraWR [23] | $0.3546 \pm 0.0105$ | $0.1905 \pm 0.0188$ | $0.3544 \pm 0.0121$ |
| CNN-Transformer [55] | $0.3678 \pm 0.0078$ | $0.2072 \pm 0.0183$ | $0.3642 \pm 0.0088$ |
| FFCL [6] | $0.3641 \pm 0.0092$ | $0.2078 \pm 0.0201$ | $0.3645 \pm 0.0132$ |
| ST-Transformer [8] | $0.3052 \pm 0.0072$ | $0.1083 \pm 0.0121$ | $0.2833 \pm 0.0105$ |
| BIOT [NeurIPS23] [17] | $0.3837 \pm 0.0187$ | $0.2261 \pm 0.0262$ | $0.3856 \pm 0.0203$ |
| LaBraM [ICLR24] [18] | $0.3976 \pm 0.0138$ | $0.2386 \pm 0.0209$ | $0.3974 \pm 0.0111$ |
| CBraMod [ICLR25] [21] | $0.4091 \pm 0.0097$ | $0.2569 \pm 0.0143$ | $0.4101 \pm 0.0108$ |
| **NEURIPT (Ours)** | $\mathbf{0.4104} \pm 0.0021$ | $\mathbf{0.2629} \pm 0.0039$ | $\mathbf{0.4158} \pm 0.0037$ |

**Preprocessing**   EEG signals were band-pass filtered between 0.1 Hz and 30 Hz and downsampled to 64 Hz. A 50 Hz notch filter was applied to remove power line interference. Each trial was segmented into non-overlapping 1-second epochs, resulting in a total of 115,001 samples. Following CBraMod [21], data from subjects 1–10 were used for training, while subjects 11–15 were reserved for testing. Detailed results are reported in Table 22.

**Experimental Configuration**   Each 1-second segment was treated as an independent sample labeled according to the corresponding emotion category. The model was trained for 10 epochs using a batch size of 256. We use both bipolar and non-bipolar methods, and the non-bipolar one is slightly better than the bipolar one.

**Results** Our proposed NEURIPT model achieves state-of-the-art performance on SEED-V emotion recognition task, outperforming existing models across the evaluation metrics. These results highlight the effectiveness and robustness of our approach in modeling complex emotional dynamics from EEG signals.

### E.2.7 Motor Imagery Task: BCIC-IV-2A (4-class)

The BCIC-IV-2A dataset [4] comprises EEG recordings from 9 healthy subjects performing 4 motor-imagery tasks: left hand (Class 1), right hand (Class 2), feet (Class 3) and tongue (Class 4). Each subject completed two sessions on separate days; each session contains six runs of 48 trials (12 trials per class), yielding 288 trials per session. Signals were acquired at 250 Hz from 22 channels positioned according to the international 10–20 system and band-pass filtered between 0.5 Hz and 100 Hz. We also present the analysis of expert participation on the this dataset in Figure 6, and report the Pearson correlation between class logits and channel perturbations induced by Gaussian multiplicative noise in Figure 4. The channel relationships are visualized in Figure 8.

Table 23: The results of different methods on motor imagery classification (BCIC-IV-2A, 4-class).

| Methods | Balanced Accuracy | Cohen's Kappa | Weight F1 |
|---|---|---|---|
| EEGNet [52] | $0.4482 \pm 0.0094$ | $0.2693 \pm 0.0121$ | $0.4226 \pm 0.0108$ |
| EEGConformer [53] | $0.4696 \pm 0.0106$ | $0.2924 \pm 0.0141$ | $0.4533 \pm 0.0128$ |
| SPaRCNet [54] | $0.4635 \pm 0.0117$ | $0.2847 \pm 0.0147$ | $0.4432 \pm 0.0126$ |
| ContraWR [23] | $0.4678 \pm 0.0125$ | $0.2905 \pm 0.0160$ | $0.4413 \pm 0.0142$ |
| CNN-Transformer [55] | $0.4600 \pm 0.0108$ | $0.2800 \pm 0.0148$ | $0.4460 \pm 0.0114$ |
| FFCL [6] | $0.4470 \pm 0.0143$ | $0.2627 \pm 0.0176$ | $0.4238 \pm 0.0139$ |
| ST-Transformer [8] | $0.4575 \pm 0.0145$ | $0.2733 \pm 0.0198$ | $0.4471 \pm 0.0142$ |
| BIOT [NeurIPS23] [17] | $0.4748 \pm 0.0093$ | $0.2997 \pm 0.0139$ | $0.4607 \pm 0.0125$ |
| LaBraM [ICLR24] [18] | $0.4869 \pm 0.0085$ | $0.3159 \pm 0.0154$ | $0.4758 \pm 0.0103$ |
| CBraMod [ICLR25] [21] | $0.5138 \pm 0.0066$ | $0.3518 \pm 0.0094$ | $0.4984 \pm 0.0085$ |
| **NEURIPT (Ours)** | $\mathbf{0.5504} \pm 0.0072$ | $\mathbf{0.4005} \pm 0.0121$ | $\mathbf{0.5376} \pm 0.0086$ |

**Preprocessing** Raw EEG signals were band-pass filtered between 0.1 Hz and 30 Hz, followed by downsampling to 64 Hz. For each trial, we extracted the segment from 2 s to 6 s after cue onset and assigned labels according to the instructed motor imagery class. This procedure yielded 288 non-overlapping 4 s segments per session. Signals were re-referenced using 16 predefined bipolar channel pairs. The model input comprised the 16 bipolar channels. Following CBraMod [21], data from subjects 1–5 were used for training, subjects 6–7 for validation, and subjects 8–9 for testing. The detailed channel configuration and corresponding sample counts are provided in Table 23.

**Experimental Configuration** Optimization employed a cosine-annealed learning rate schedule, warming up linearly from $1.7 \times 10^{-5}$ to $3.5 \times 10^{-5}$, then decaying back to $1.7 \times 10^{-5}$ by epoch 150. We trained with a batch size of 16 for 150 epochs.

**Results** Our proposed NEURIPT model achieves state-of-the-art performance on BCIC-IV-2A motor imagery classification, outperforming existing baselines across all evaluation metrics. These results demonstrate the effectiveness and robustness of our approach.

### E.2.8 Abnormal Detection: TUAB (2-class) and Event Type Classification: TUEV (6-class)

The TUEV dataset [2] is a curated subset of the Temple University Hospital EEG Corpus (TUEG), enriched with expert annotations for epileptiform and non-epileptiform events. Each EEG segment is labeled into one of six categories: spikes and sharp waves (Class 1), generalized periodic epileptiform discharges (Class 2), periodic unilateral epileptiform discharges (Class 3), eye movements (Class 4), artifacts (Class 5), and background (Class 6). Recordings were acquired using 23 EEG channels with a sampling rate of 250 Hz. We present the analysis of expert participation on this dataset in Figure 15. The channel relationships are visualized in Figure 8.

The TUAB dataset [2] is a large-scale EEG corpus annotated for abnormality detection. Each recording is labeled as either normal or abnormal based on clinical reports. Similar to TUEV, the EEG signals were recorded using 23 channels with a sampling rate of 250 Hz. The dataset serves as a benchmark for evaluating automated EEG abnormality detection methods. We present the analysis of expert participation on this dataset in Figure 9. The channel relationships are visualized in Figure 8.

Table 24: Comparison of different methods on TUAB (2-class) and TUEV (6-class).

| Method | TUAB | | | TUEV | | |
|---|---|---|---|---|---|---|
| | Balanced Acc. | AUC-PR | AUROC | Balanced Acc. | Cohen's Kappa | Weighted F1 |
| EEGNet [52] | $0.7642 \pm 0.0036$ | $0.8299 \pm 0.0043$ | $0.8412 \pm 0.0031$ | $0.3876 \pm 0.0143$ | $0.3577 \pm 0.0155$ | $0.6539 \pm 0.0120$ |
| EEGConformer [53] | $0.7758 \pm 0.0049$ | $0.8427 \pm 0.0054$ | $0.8445 \pm 0.0038$ | $0.4074 \pm 0.0164$ | $0.3967 \pm 0.0195$ | $0.6983 \pm 0.0152$ |
| SPaRCNet [54] | $0.7896 \pm 0.0018$ | $0.8414 \pm 0.0018$ | $0.8676 \pm 0.0012$ | $0.4161 \pm 0.0262$ | $0.4233 \pm 0.0181$ | $0.7024 \pm 0.0104$ |
| ContraWR [23] | $0.7746 \pm 0.0041$ | $0.8421 \pm 0.0104$ | $0.8456 \pm 0.0074$ | $0.4384 \pm 0.0349$ | $0.3912 \pm 0.0237$ | $0.6893 \pm 0.0136$ |
| CNN-Transformer [55] | $0.7777 \pm 0.0022$ | $0.8433 \pm 0.0039$ | $0.8461 \pm 0.0013$ | $0.4087 \pm 0.0161$ | $0.3815 \pm 0.0134$ | $0.6854 \pm 0.0293$ |
| FFCL [6] | $0.7848 \pm 0.0038$ | $0.8448 \pm 0.0065$ | $0.8569 \pm 0.0051$ | $0.3979 \pm 0.0104$ | $0.3732 \pm 0.0188$ | $0.6783 \pm 0.0120$ |
| ST-Transformer [8] | $0.7966 \pm 0.0023$ | $0.8521 \pm 0.0026$ | $0.8707 \pm 0.0019$ | $0.3984 \pm 0.0228$ | $0.3765 \pm 0.0306$ | $0.6823 \pm 0.0190$ |
| BIOT[NeurIPS23] [17] | $0.7959 \pm 0.0057$ | $0.8792 \pm 0.0023$ | $0.8815 \pm 0.0043$ | $0.5281 \pm 0.0225$ | $0.5273 \pm 0.0249$ | $0.7492 \pm 0.0082$ |
| LaBraM-Huge[ICLR24] [18] | $0.8258 \pm 0.0011$ | $0.9204 \pm 0.0011$ | $0.9162 \pm 0.0016$ | $0.6616 \pm 0.0170$ | $0.6745 \pm 0.0195$ | $0.8329 \pm 0.0086$ |
| EEGPT[NeurIPS24] [20] | $0.7983 \pm 0.0030$ | - | $0.8718 \pm 0.8718$ | $0.6232 \pm 0.0114$ | $0.6351 \pm 0.0134$ | $0.8187 \pm 0.0063$ |
| NeuroLM-XL[ICLR25] [19] | $0.7969 \pm 0.0091$ | $0.7219 \pm 0.0082$ | $0.7884 \pm 0.0194$ | $0.4679 \pm 0.0356$ | $0.4570 \pm 0.0498$ | $0.7359 \pm 0.0219$ |
| CBraMod[ICLR25] [21] | $0.8289 \pm 0.0022$ | $\mathbf{0.9258 \pm 0.0008}$ | $\mathbf{0.9227 \pm 0.0011}$ | $0.6671 \pm 0.0107$ | $0.6772 \pm 0.0096$ | $0.8342 \pm 0.0064$ |
| **NEURIPT (Ours)** | $\mathbf{0.8293 \pm 0.0016}$ | $0.9040 \pm 0.0022$ | $0.8949 \pm 0.0021$ | $\mathbf{0.6761 \pm 0.0133}$ | $\mathbf{0.6970 \pm 0.0185}$ | $\mathbf{0.8428 \pm 0.0089}$ |

**Preprocessing**  All EEG recordings were band-pass filtered between 0.1 Hz and 30 Hz to attenuate low- and high-frequency noise, and a 60 Hz notch filter was applied to remove power-line interference. Signals were then resampled to 64 Hz and segmented into fixed-length epochs: 5 s non-overlapping segments for TUEV, and 10 s segments for TUAB. We maintained the original training/test partitions provided by each dataset to ensure a fair comparison with the three baselines-LaBraM[18], EEGPT [20] and CBraMod[21]. The training subjects were further divided into training and validation subsets in an 80% / 20% ratio.

**Experimental Configuration**  For TUEV, each 5-second segment was treated as an independent sample labeled according to the corresponding category. The model was trained for 20 epochs using a batch size of 16. The learning rate peaked at $1 \times 10^{-5}$, and then gradually decayed to a minimum of $1 \times 10^{-7}$. For TUAB, each 10-second segment was treated as an independent sample labeled according to the corresponding category. The model was trained for 10 epochs using a batch size of 16. The learning rate peaked at $5 \times 10^{-6}$, and then gradually decayed to a minimum of $3 \times 10^{-7}$.

**Results**  Our proposed NEURIPT model achieves state-of-the-art performance on the TUEV dataset, consistently outperforming existing baselines across all evaluation metrics. These results highlight the effectiveness and robustness of our method. On the TUAB abnormality detection task, NEURIPT attains higher balanced accuracy compared to prior methods. While the performance on the other two metrics is slightly lower than the best results, it still exceeds most baseline models. This indicates that NEURIPT is both competitive and effective, demonstrating strong generalization ability in learning transferable EEG representations.

# F Discussion

## F.1 Limitation

While NEURIPT incorporated several important characteristics of EEG signals into its neural architecture to learn more generalizable representations, there remain several open challenges and opportunities for future work. (1) For instance, additional EEG-specific aspects, such as brain connectivity representations, have not yet been fully explored and could enhance decoding performance. (2) Moreover, as with many foundation models, NEURIPT requires a large number of parameters, resulting in increased memory usage and higher computational costs during both training and inference compared to non-foundation models. (3) Our ability to further scale the model and investigate scaling laws was constrained by the limited GPU resources available. (4) Lastly, although NEURIPT achieved SOTA performance on average across multiple datasets, there is still room to improve neural decoding accuracy to fully meet the practical demands of BCI systems for real-world and clinical applications.

## F.2 Broader Impacts

This work advances neural decoding for general EEG-based neural interfaces, with potential benefits across a wide range of BCI applications, including clinical diagnosis, emotion recognition, and user intent control. These advancements have broader implications for enhancing human well-being by expanding access to brain–computer interface (BCI) technologies and enabling more diverse real-world applications. While our method is primarily developed for foundation models (FMs), it is designed to leverage key EEG characteristics, making it also adaptable to task- and setup-specific models for a wide range of application-specific scenarios. Furthermore, our findings provide valuable insights for the development of more advanced foundation models that incorporate other EEG-specific characteristics, such as brain connectivity.

## F.3 Potential Misuse

As EEG foundation models become increasingly powerful for healthcare and brain–computer interface (BCI) applications, it is crucial to acknowledge their potential for misuse, for example, unauthorized cognitive surveillance or profiling. The ability to infer mental states or cognitive intent, if applied without proper oversight, raises profound ethical concerns. Addressing these risks will require the development of neurotechnology regulations, data protection policies, and ethics frameworks to ensure responsible use and safeguard individual neural privacy.

# G Further Visualization

Further visualizations start on the next page.

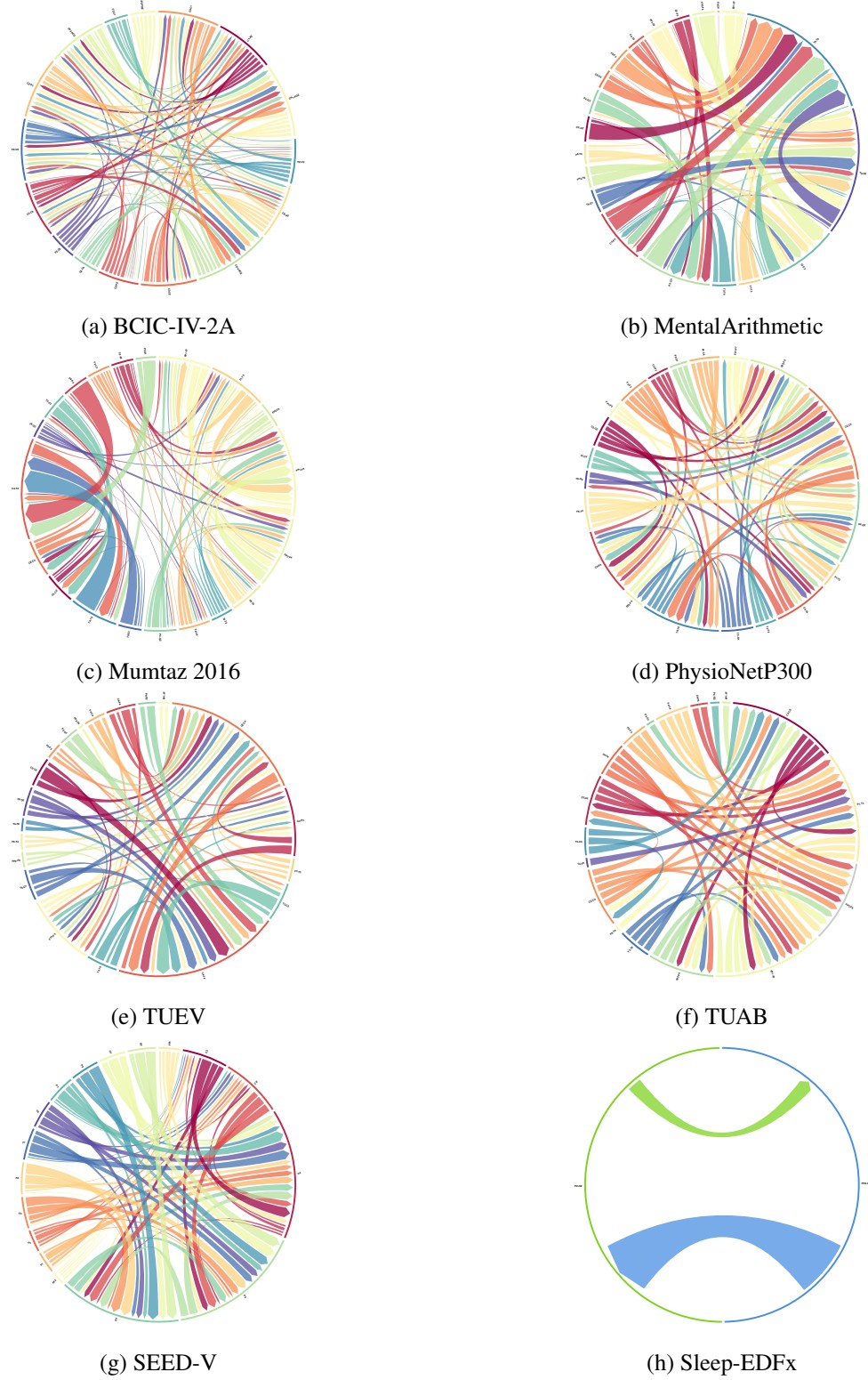

Figure 8: Inter-channel relationships for eight downstream tasks.

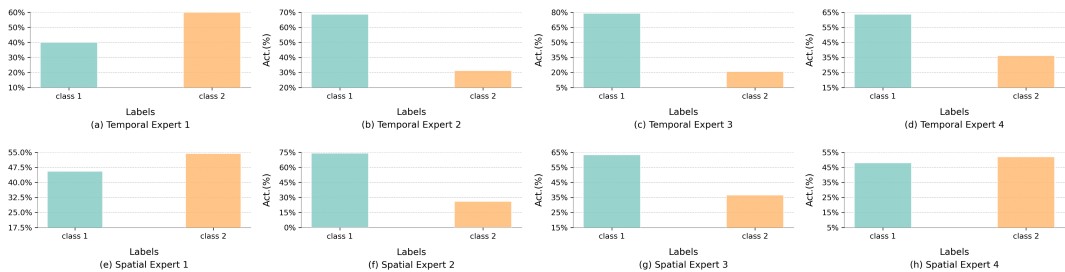

Figure 9: Analysis of expert participation on the TUAB dataset.

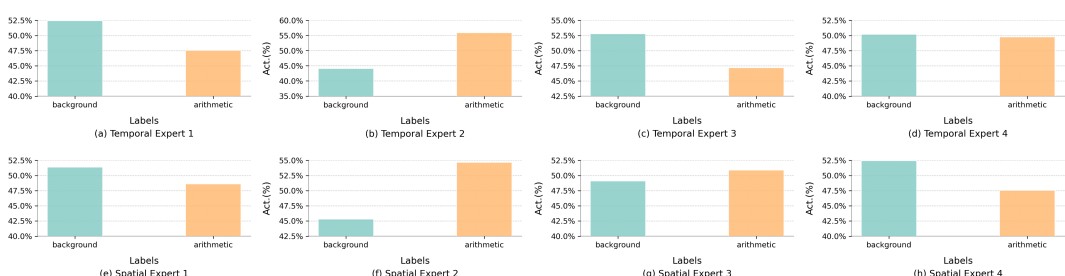

Figure 10: Analysis of expert participation on the MentalArithmetic dataset.

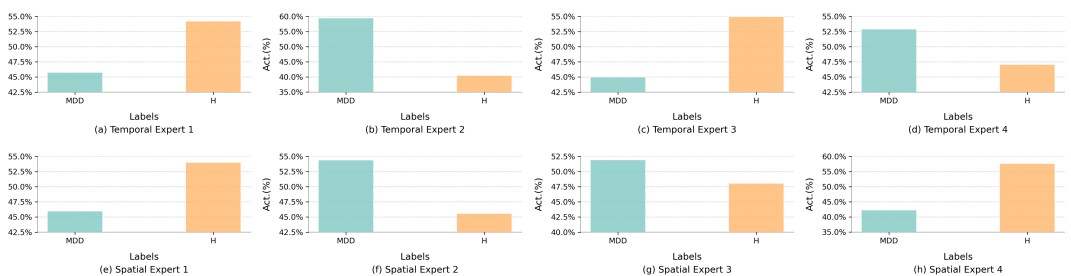

Figure 11: Analysis of expert participation on the Mumtaz dataset.

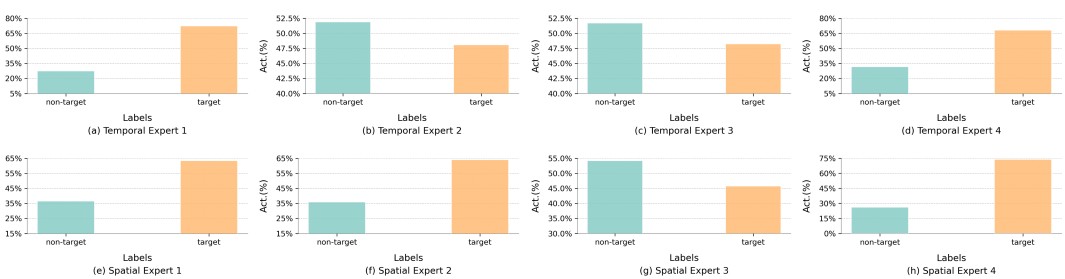

Figure 12: Analysis of expert participation on the P300 dataset.

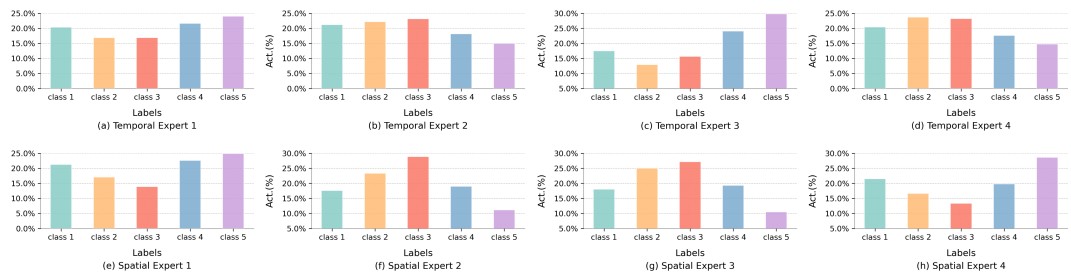

Figure 13: Analysis of expert participation on the SleepEDF dataset.

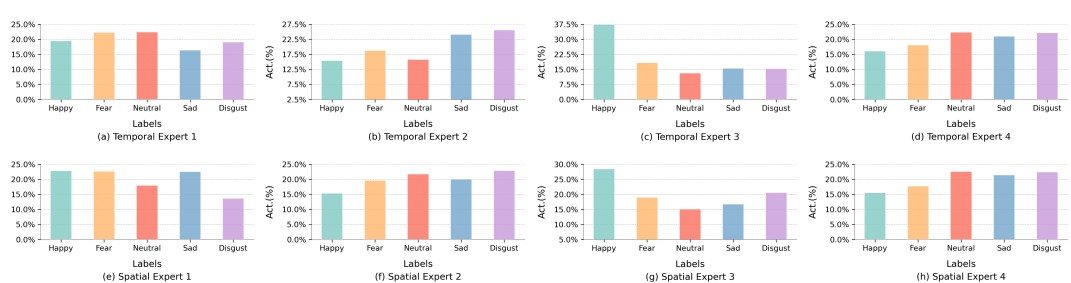

Figure 14: Analysis of expert participation on the SEED-V dataset.

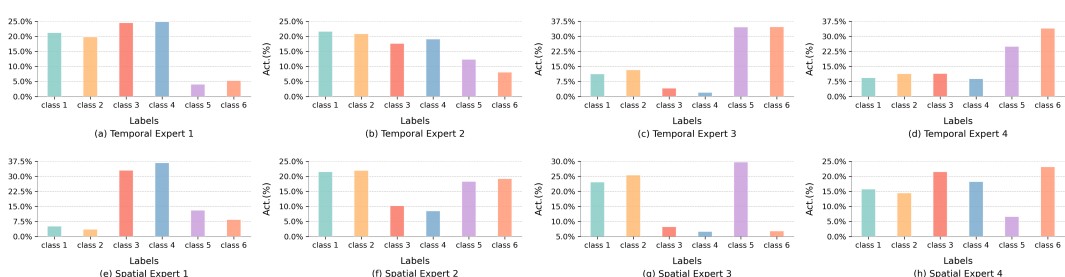

Figure 15: Analysis of expert participation on the TUEV dataset.

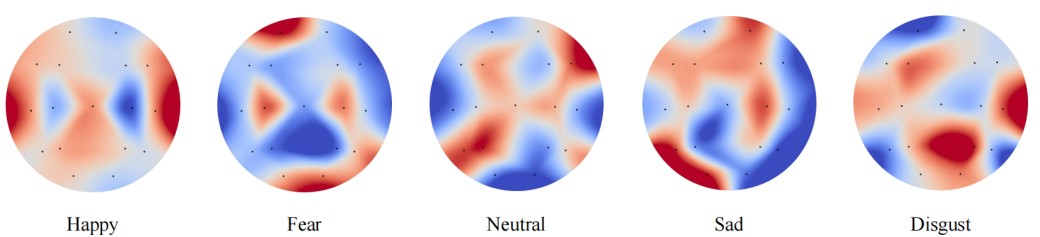

Figure 16: Pearson correlation between class logits and channel perturbation using Gaussian multiplicative noise on SEED-V.

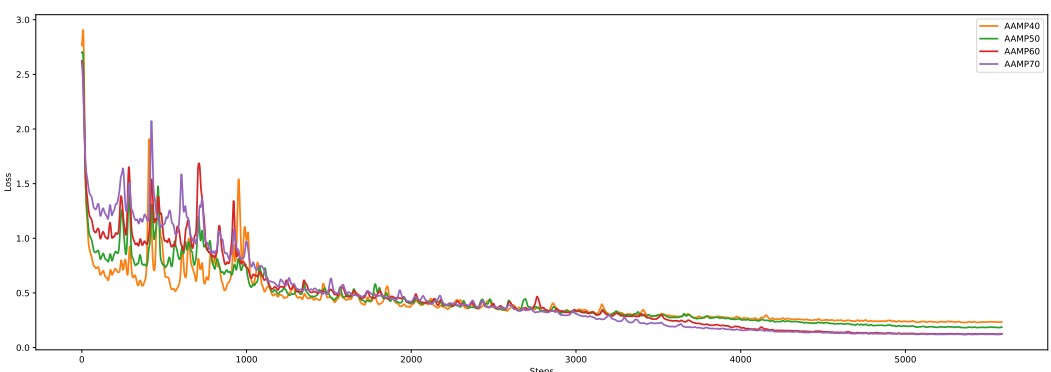

Figure 17: Loss with different mask ratios

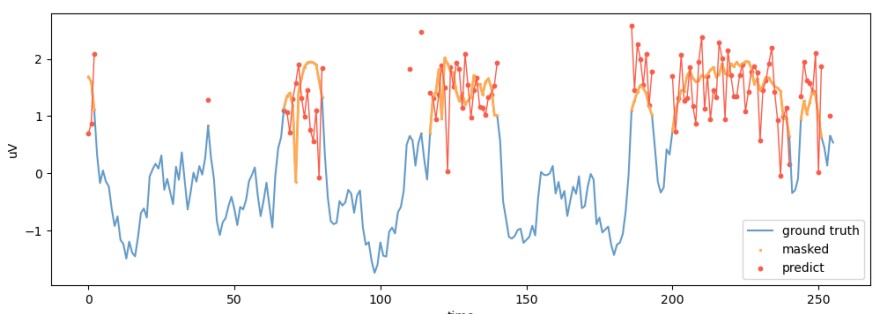

Figure 18: Visualization of AAMP based reconstruction (a).

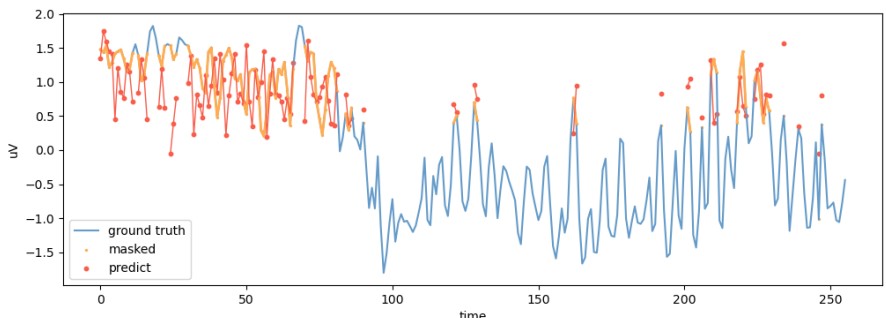

Figure 19: Visualization of AAMP based reconstruction (b).

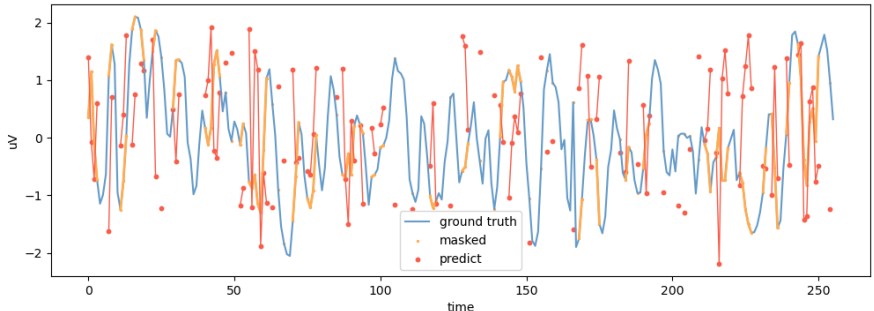

Figure 20: Visualization of reconstruction with BERT-style masking.

