# OpenReview forum: "NeurIPT: Foundation Model for Neural Interfaces"
_NeurIPS.cc/2025/Conference — NeurIPS 2025 poster_

### Official Review · Reviewer_LoU3 · 2025-06-11

**Clarity:** 3
**Significance:** 3
**Originality:** 3
**Rating:** 4
**Confidence:** 4

**Summary:**

This paper proposes a foundation model, NeurIPT, developed for EEG-based neural interfaces. NeurIPT could capture both homogeneous and heterogeneous spatio-temporal characteristics inherent in EEG signals. The designed techniques like Amplitude-Aware Masked Pretraining, PMoE are impressive and well-motivated. Sufficient experiments and analysis are conducted to demonstrate the effectiveness of the model.

**Questions:**

* What is the numerical range of spatial coordinates?
* What is the difference between amplitude-aware masking and random patch masking? A larger patch may avoid interpolation.
* Are the experimental settings in downstream tasks cross-subject, i.e., the test subjects are excluded from the fine-tuning data? It is crucial in tasks like mental disorders diagnosis.
* Does IILP gathers the representations from all encoder layers? What is the motivation behind this? Most of the methods only use the representations from the last encoder layer.
* Why the Top k in MoE sets to 0.5?

**Ethical Concerns:**

["NO or VERY MINOR ethics concerns only"]

**Final Justification:**

The ablation analysis and some clarification issues are addressed in rebuttal.

**Limitations:**

yes

**Paper Formatting Concerns:**

no paper formatting concerns

**Quality:**

3

**Strengths And Weaknesses:**

strength

* The designed techniques are well-motivated.
* The downstream tasks are sufficient.

weakness

* No information about the statistical significance of the experiments (e.g., error bars, p-value) is provided.
* Some crucial information in experimental settings is missed.
* The analysis experiments are insufficient. The analysis of different model components should be conducted on all downstream datasets to varify the effectiveness. The results on only one or two datasets are not convincing.
* The IILP pooling strategy should be compared with other pooling methods like generalized mean pooling.

---

> ### Author Rebuttal · Authors · 2025-07-31
>
> **Dear reviewer LoU3:**
>
> **We appreciate your insightful comments. Below, we respond to each of your concerns.**
>
> ---
>
> > [W1] No information about the statistical significance of the experiments.
>
> [A1] We had presented comprehensive details for each downstream task, including statistical metrics (mean, standard deviation), in Tables 11–17 of Appendix E.2 (Downstream Datasets) in the supplementary materials, as we could not include them in the main text due to space constraints and our extensive experiments. Consistent with our baselines (e.g., LaBraM, CBraMod), all reported results are averages computed over five independent runs with different random seeds, accompanied by standard deviation values to facilitate meaningful and rigorous comparisons.
>
> &nbsp;
>
> > [W2] Some experimental settings are missed.
>
> [A2] We respectfully clarify that detailed information on the experimental settings was comprehensively provided across both the main text and the appendix. Should any additional details or clarifications be required, please do not hesitate to let us know if you cannot find any setting.
>
> Specifically, Sections 3.1 (Pre-training) and 3.2 (Downstream BCI Tasks) introduce the datasets, preprocessing methods, experiment configurations, and evaluation metrics. Additionally, Appendix D (Hyperparameter and Implementation Details) supplements with parameter choices and implementation specifics. Furthermore, Section E.1.2 and Table 10 elaborate on the filtering process and statistics of the pre-training dataset, while Sections E.2.2–E.2.8 deliver extensive descriptions of each downstream dataset. For complete reproducibility, the experimental scripts and hyperparameter configurations are accessible in the directory *./scripts/*, and all preprocessing steps are transparently documented in the folders *./preprocessing/pretrain/* and *./preprocessing/downstream/*. The provided code fully aligns with the manuscript's contents.
>
> We sincerely apologize for any inconvenience this may have caused, and kindly note that the complete appendix and source code are included within the Supplementary Material zip file.
>
> &nbsp;
>
> > [W3] Insufficient analysis experiments of different model components.
>
> [A3] We have expanded our ablation analyses across more downstream datasets. Specifically, comprehensive experiments for the component IILP have been detailed in [A4]. Below, we provide comparisons focusing on the AAMP, 3D positional encoding, and the PMoE strategies.
>
> |Masking Strategies / Datasets|TUEV|MentalArithmetic|Mumtaz2016|SEED-V|PhysioP300|Sleep-EDFx|
> |:-:|:-:|:-:|:-:|:-:|:-:|:-:|
> |Random Masking|67.35|84.38|97.14|33.89|65.64|69.68|
> |AAMP (Ours)|**67.61**|**86.46**|**98.03**|**41.04**|**67.31**|**70.47**|
>
> These results demonstrate consistent improvements of AAMP across multiple downstream datasets, demonstrating the effectiveness and generalization capability of our proposed masking strategy.
>
> |Encoding Strategies / Datasets|TUEV|MentalArithmetic|Mumtaz2016|SEED-V|
> |:-:|:-:|:-:|:-:|:-:|
> |[1]|67.72|74.65|96.56|35.03|
> |[2]|64.78|71.53|96.63|35.52|
> |[3]|63.81|71.52|95.56|37.40|
> |IILP (Ours)|**68.94**|**75.69**|**97.07**|**39.34**|
>
> alternative methods are included:
>
> - [1] trigonometric functions employed by vanilla Transformer,
> - [2] 1D learnable embeddings,
> - [3] 2D learnable embeddings introduced in LaBraM,
>
> The results above clearly demonstrate that our proposed 3D electrode embedding achieves consistently superior performance across multiple datasets, substantiating the notable advantage of explicitly leveraging physical electrode positions.
>
> |MoE Strategies / Datasets|TUEV|MentalArithmetic|Mumtaz2016|SEED-V|PhysioP300|BCIC-2A|
> |:-:|:-:|:-:|:-:|:-:|:-:|:-:|
> |w/o Expert|65.83|72.92|93.41|39.14|64.53|44.44|
> |Uniform|65.91|70.49|95.00|39.33|65.66|41.93|
> |Shrinking|65.80|73.96|93.08|39.21|65.99|**44.62**|
> |Progressive (Ours)|**68.94**|**75.69**|**97.07**|**39.34**|**66.58**|44.01|
>
> We have expanded Table 3 by evaluating different MoE strategies across multiple datasets, demonstrating the robustness of our PMoE approach. Furthermore, we have also empirically compared other alternative and gradual configurations:
>
> |PMoE Strategies / Datasets|TUEV|MentalArithmetic|Mumtaz2016|SEED-V|PhysioP300|BCIC-2A|
> |:-:|:-:|:-:|:-:|:-:|:-:|:-:|
> |[0,0,2,4,4,6]|**68.94**|75.69|97.07|39.34|66.58|**44.01**|
> |[0,0,2,3,4,5]|67.87|**76.39**|96.81|**39.88**|66.34|41.58|
> |[0,0,2,4,6,8]|66.41|75.69|96.57|39.11|**67.70**|41.15|
> |[0,0,3,6,9,12]|66.08|74.83|**97.15**|39.35|66.63|38.19|
>
> These alternative patterns also yield competitive performance, suggesting that the effectiveness of the PMoE approach stems from its inherent progressive strategy, rather than relying on any specific expert allocation, demonstrating robustness across diverse progressive configurations.
>
> &nbsp;
>
> > [W4] IILP pooling strategy should be compared with alternative methods.
>
> [A4] We have expanded Table 4 (IILP ablation) by incorporating comparisons between IILP and alternative classification methods, including mean pooling across multiple datasets:
>
> |Pooling Strategies / Datasets|TUEV|MentalArithmetic|Mumtaz2016|SEED-V|PhysioP300|BCIC-2A|
> |:-:|:-:|:-:|:-:|:-:|:-:|:-:|
> |w/o Pooling|62.33|75.69|78.21|38.90|**67.82**|45.14|
> |Mean Pooling|64.74|79.51|96.22|37.62|66.72|37.24|
> |Hemispheres|64.45|81.94|97.82|39.22|67.11|43.49|
> |Coronal|68.77|73.26|96.99|39.35|66.98|43.75|
> |Sagittal|67.21|80.21|91.41|39.42|65.66|45.31|
> |IILP (Ours)|**68.94**|**86.46**|**98.03**|**41.04**|67.31|**55.04**|
>
> The above results further demonstrate the consistent superiority of our approach.
>
> &nbsp;
>
> > [Q1] Numerical range of spatial coordinates.
>
> [A5] The spatial coordinate ranges (in $mm$) of the two EEG standard systems involved in downstream tasks are as follows:
>
> - EEG 10-20 system: $x\in[-86.076,86.169],\quad y\in[-118.565,88.247],\quad z\in[-68.031,100.244]$
> - EEG 10-05 system: $x\in[-87.362,88.625],\quad y\in[-119.343,88.247],\quad z\in[-68.031, 101.269]$
>
> Specifically, the coordinates correspond to the fsaverage MRI coordinate system, which is identical to the Montreal Neurological Institute (MNI) coordinate framework, with the origin located at the anterior commissure. These coordinates are extracted from the widely used MNE-Python library.
>
> &nbsp;
>
> > [Q2] Difference between amplitude-aware masking and random patch masking, and impact of patch size
>
> [A6] Our amplitude-aware masking is a context-aware strategy that encourages the model to learn signal-specific characteristics and structural patterns, with amplitude serving as a proxy for signal energy. This approach reduces the risk of trivial reconstruction via simple interpolation methods (e.g., linear), which models might otherwise exploit.
>
> We appreciate this suggestion regarding the impact of larger patch sizes, which could mitigate trivial interpolation issues inherent in random masking. However, increasing the patch size substantially escalates the complexity faced by the embedding layers, typically implemented as linear transformations. This issue could impose a significant burden on the model's capacity to effectively capture detailed local signal characteristics, thereby negatively affecting downstream performance.
>
> For instance, in CV, ViT [1] demonstrated notable performance improvements when reducing the patch size from 32 to 16, and subsequent studies [2] found further gains by decreasing the patch size even further down to 1. Similarly, in NLP, mainstream models benefit from fine-grained, byte-level or sub-word-level embeddings.
>
> In contrast, our proposed amplitude-aware masking strategy (AAMP) intrinsically circumvents the performance-interpolation trade-off, directly guiding the model to learn meaningful signal representations. Empirical evidence supporting our claim has been further validated by experiments in [A3], reinforcing the practical effectiveness of AAMP compared to conventional random masking.
>
> &nbsp;
>
> > [Q3] Are downstream tasks cross-subject?
>
> [A7] Yes, all downstream datasets used in our study adopt a cross-subject evaluation setting, where the test subjects are strictly excluded from the fine-tuning process. This setting is consistent with other baselines as well.
>
> &nbsp;
>
> > [Q4] Does IILP gather all layers? Why?
>
> [A8] Yes, IILP aggregates representations from all layers rather than only the last one. This design choice leverages the hierarchical architecture of our backbone model, which inherently captures EEG signal features at multiple granularities. By combining fine-grained details from early layers with broader-level representations from deeper layers, IILP enriches the final representation, offering a comprehensive, multi-scale perspective beneficial for the downstream classification tasks.
>
> &nbsp;
>
> > [Q5] Why sets topk to 0.5?
>
> [A9] The choice of setting Top-k to 0.5 was a default design decision rather than the outcome of an explicit hyperparameter tuning process. We maintained this default setting uniformly across all downstream tasks and realized it worked satisfactorily without the need for tuning. Nevertheless, we have made this parameter available as a configurable hyperparameter, allowing flexibility for adjustment according to specific downstream requirements or datasets.
>
> &nbsp;
>
> References:
>
> [1] An Image is Worth 16x16 Words: Transformers for Image Recognition at Scale (ICLR 2021)
>
> [2] An Image is Worth More Than 16x16 Patches: Exploring Transformers on Individual Pixels (ICLR 2025)
>
> ---
>
> **Thank you again for your valuable comments. We hope our responses have addressed your concerns and strengthened your impression of our work. We remain available throughout the rebuttal period to clarify any further questions you may have. Looking forward to hearing from you :-) !**

---

> > ### Comment · Reviewer_LoU3 · 2025-08-02
> >
> > Thanks for your response. Most issues are solved, and I decide to keep my original score. I hope those clarifications and experiments could be integrated in the final version.

---

> ### Author Response · Authors · 2025-08-02
> **Thanks for your timely reply and the kind words**
>
> **Thank you for confirming that our responses have adequately addressed your concerns and for maintaining your positive assessment.**
>
> **We sincerely appreciate your professional and constructive feedback, which has significantly enhanced the clarity and quality of our manuscript. As recommended, we will integrate all clarifications and additional experimental results into the final version.**
>
> **Should you have any further comments or suggestions, please do not hesitate to reach out. We remain available throughout the revision process. Thank you once again for your valuable input.**

---

### Official Review · Reviewer_L8YY · 2025-06-30

**Clarity:** 3
**Significance:** 3
**Originality:** 3
**Rating:** 4
**Confidence:** 5

**Summary:**

This paper presents NeurIPT , a new foundation model for EEG-based neural interfaces. It addresses key challenges in generalizing across diverse EEG data — such as varying subjects, tasks, and electrode setups — by introducing an amplitude-aware masked pretraining strategy and a progressive Mixture-of-Experts architecture. The model also incorporates 3D electrode coordinates and a novel pooling mechanism to better capture spatial brain patterns. Evaluated across eight BCI datasets, NeurIPT achieves strong results under fine-tuning, showing promising generalization and broad applicability in EEG modeling.

**Questions:**

- The paper adopts several training techniques commonly used in large language models, such as the SwiGLU activation function. While these components have shown benefits in NLP tasks, it is unclear whether they bring measurable improvements to EEG foundation model training compared to more traditional choices like ReLU or GELU. Some ablation or discussion on this would be valuable.
- The authors mention that additional experimental results and details are included in the appendix, but it is not attached directly to the main manuscript. Instead, it appears to be submitted as a separate file. This format makes it less convenient for reviewers and readers to access supplementary material. Why not follow the standard NeurIPS practice of appending it directly after the main text?

**Ethical Concerns:**

["NO or VERY MINOR ethics concerns only"]

**Final Justification:**

The author's response addressed most of the reviewer's concerns, and the reviewer decided to maintain the positive rating while raising the clarity score to 3.

**Limitations:**

yes

**Quality:**

3

**Strengths And Weaknesses:**

### Strengths

- The proposed Amplitude-Aware Masked Pretraining (AAMP) introduces a novel perspective on how to design pretraining objectives for EEG foundation models. It moves beyond traditional random masking by focusing on signal amplitude, which could inspire future work in task-specific representation learning.

- The introduction of Progressive Mixture-of-Experts (PMoE) is conceptually interesting and offers a promising way to model diverse temporal patterns in EEG signals by gradually increasing the specialization of expert subnetworks.

- The use of 3D physical coordinates of electrodes for embedding provides a more structured and biologically grounded approach to channel encoding, which may help improve cross-dataset generalization.

- The model was evaluated across a wide range of downstream BCI tasks, including motor imagery, emotion recognition, and seizure detection. This variety adds credibility to the claim that NeurIPT has broad applicability.


### Weaknesses

- While the use of 3D electrode coordinates is a nice design choice, the paper does not clearly show how this compares with other existing spatial encoding strategies, such as those used in LaBraM or other EEG foundation models. A direct comparison or ablation would help clarify its added value.

- The authors adopt Crossformer as the backbone architecture but do not provide sufficient discussion or analysis regarding its advantages over standard Transformer architectures in the context of EEG modeling. This leaves an important question about architectural choice unaddressed.

- Although the ablation study shows that AAMP leads to better performance on downstream classification tasks, it's unclear why that is the case. In particular, the authors note that random masking results in lower reconstruction loss — suggesting better interpolation ability — yet performs worse in classification. A deeper analysis of *why* AAMP helps classification would strengthen the contribution.

- Similarly, while the PMoE mechanism appears beneficial, the link between expert activation patterns and actual performance gains remains speculative. The observation that different classes activate different experts is interesting, but it's not shown whether this is a cause or a side effect of improved accuracy.

- The paper reports that IILP outperforms other pooling strategies based on anatomical brain region groupings such as Coronal, Sagittal, and Hemispheric divisions. However, the reason for this improvement is not clearly explained — is it due to better functional alignment, more balanced regional aggregation, or something else?

- Some minor issues include the lack of clarity around certain experimental details — for instance, Table 5 does not specify which dataset the results are based on. Additionally, the paper lacks information on training time and computational resources, which are important for reproducibility and assessing feasibility.

---

> ### Author Rebuttal · Authors · 2025-07-31
>
> **Dear reviewer L8YY:**
>
> **We appreciate your insightful comments. Below, we respond to each of your concerns.**
>
> ---
> > [W1] Ablation of 3D electrode coordinates.
>
> [A1] To rigorously examine the benefit of using 3D electrode coordinates, we conducted a comprehensive ablation study comparing against several existing spatial encoding methods:
>
> - [1] trigonometric functions employed by vanilla Transformer,
> - [2] 1D learnable embeddings,
> - [3] 2D learnable embeddings introduced in LaBraM,
>
> |Encoding Strategies / Datasets|TUEV|MentalArithmetic|Mumtaz2016|SEED-V|
> |:-:|:-:|:-:|:-:|:-:|
> |[1]|67.72|74.65|96.56|35.03|
> |[2]|64.78|71.53|96.63|35.52|
> |[3]|63.81|71.52|95.56|37.40|
> |IILP (Ours)|**68.94**|**75.69**|**97.07**|**39.34**|
>
> The results above demonstrate the effectiveness of our proposed 3D electrode embedding across multiple datasets, substantiating the advantage of explicitly leveraging physical electrode positions.
>
> &nbsp;
>
> > [W2] Reasons for choosing Crossformer as the backbone architecture.
>
> [A2] Thank you for requesting clarification on our backbone. We have now revised lines 181-185 and 582-583 to explicitly address the rationale behind selecting Crossformer [1].
>
> Specifically, we build upon the Two-Stage Attention (TSA) from Crossformer [1], which sequentially performs cross-time and cross-dimension computations with a hierarchical architecture, and could be especially useful for capturing the intricate spatio-temporal dependencies inherent in multi-channel, multi-granularity EEG data.
>
> Moreover, TSA addresses computational efficiency compared to standard multivariate time series architectures, significantly reducing resource consumption during large-scale pre-training. Additionally, the hierarchical structure proposed in [1] naturally captures EEG signal representations at multiple granularities, providing both detailed and broader-level information. At deeper layers with coarser granularity, the reduced sequence lengths lead to lower computational demands. These advantages collectively informed our decision to adopt [1] as the backbone architecture.
>
> &nbsp;
>
> > [W3] AAMP further analysis and ablation.
>
> [A3] Our amplitude-aware masking is a context-aware strategy that encourages the model to learn signal-specific characteristics and structural patterns, with amplitude serving as a proxy for signal energy. This approach reduces the risk of trivial reconstruction via simple interpolation methods (e.g., linear), which models might otherwise exploit. Moreover, our model benefits from its multivariate design, allowing it to reconstruct masked regions by leveraging complementary, unmasked information across channels. This preserves learning effectiveness even under dense intra-channel masking conditions.
>
> Although random masking achieved lower reconstruction loss, its inferior downstream performance strongly supports our central argument: random masking inherently promotes interpolation rather than encouraging deeper structural feature learning. To further substantiate this claim, we conducted additional experiments where only the masking strategy was swapped while keeping all other model components fixed:
>
> |Masking Strategies / Datasets|TUEV|MentalArithmetic|Mumtaz2016|SEED-V|PhysioP300|Sleep-EDFx|
> |:-:|:-:|:-:|:-:|:-:|:-:|:-:|
> |Random Masking|67.35|84.38|97.14|33.89|65.64|69.68|
> |AAMP (Ours)|**67.61**|**86.46**|**98.03**|**41.04**|**67.31**|**70.47**|
>
> These results demonstrate consistent improvements of AAMP across multiple downstream datasets, demonstrating the effectiveness and generalization capability of our proposed masking strategy.
>
> &nbsp;
>
> > [W4] PMoE exploration and ablation.
>
> [A4] Thank you for acknowledging our PMoE mechanism. To address the reviewer’s concern regarding the connection between performance gains and expert activation patterns (Fig.6, 9-15), we have conducted two comprehensive ablation studies to empirically validate the effectiveness of PMoE. Specifically, we first expanded our experiments in Table 3 across multiple downstream tasks:
>
> |MoE Strategies / Datasets|TUEV|MentalArithmetic|Mumtaz2016|SEED-V|PhysioP300|BCIC-2A|
> |:-:|:-:|:-:|:-:|:-:|:-:|:-:|
> |w/o Expert|65.83|72.92|93.41|39.14|64.53|44.44|
> |Uniform|65.91|70.49|95.00|39.33|65.66|41.93|
> |Shrinking|65.80|73.96|93.08|39.21|65.99|**44.62**|
> |Progressive (Ours)|**68.94**|**75.69**|**97.07**|**39.34**|**66.58**|44.01|
>
> Results consistently demonstrate that PMoE outperforms other expert-allocation strategies. Furthermore, we explored the alternative progressive configurations:
>
> |PMoE Strategies / Datasets|TUEV|MentalArithmetic|Mumtaz2016|SEED-V|PhysioP300|BCIC-2A|
> |:-:|:-:|:-:|:-:|:-:|:-:|:-:|
> |[0,0,2,4,4,6]|**68.94**|75.69|97.07|39.34|66.58|**44.01**|
> |[0,0,2,3,4,5]|67.87|**76.39**|96.81|**39.88**|66.34|41.58|
> |[0,0,2,4,6,8]|66.41|75.69|96.57|39.11|**67.70**|41.15|
> |[0,0,3,6,9,12]|66.08|74.83|**97.15**|39.35|66.63|38.19|
>
> These alternative configurations also yield competitive performance, confirming that performance improvements indeed correlate with PMoE and its progressive configurations, rather than just being side effects. This empirical observation suggests that the effectiveness of the PMoE approach stems from its inherent progressive strategy.
>
> &nbsp;
>
> > [W5] Reasons for IILP improvement.
>
> [A5] We attribute the improvement from IILP primarily to the functional and structural alignment of brain regions, as EEG datasets are typically collected to examine different aspects of brain functionality, and brain lobes are anatomically separated by sulci.
>
> In our brain region-specific analysis (Figs. 5, 8, and 16) using attention scores, we observe consistent patterns of interaction within each brain region and clear distinctions between regions. In Fig. 5, we apply a consistent color scheme to each brain region across both the left and right panels, which we have now explicitly clarified in the revised figure caption. For instance, substantial attentional connections from the occipital lobe (red) to the right temporal lobe (yellow) indicate meaningful cross-regional interactions. Moreover, we also observe contralateral control, where voluntary movement is lateralized mainly to the motor cortex on the opposite side of the brain. For instance, the scalp heatmap on the right side of Fig. 5 highlights regional, rather than sagittal or hemispheric, patterns. Specifically, the *Left Hand* label prominently activates the right temporal lobe, whereas the *Right Hand* label activates the left temporal lobe. These observations are consistently reinforced in Figs. 8 and 16. Furthermore, we have expanded the ablation studies in Table 4 to comprehensively demonstrate IILP's comparative advantages across multiple datasets:
>
> |Pooling Strategies / Datasets|TUEV|MentalArithmetic|Mumtaz2016|SEED-V|PhysioP300|BCIC-2A|
> |:-:|:-:|:-:|:-:|:-:|:-:|:-:|
> |w/o Pooling|62.33|75.69|78.21|38.90|**67.82**|45.14|
> |Mean Pooling|64.74|79.51|96.22|37.62|66.72|37.24|
> |Hemispheres|64.45|81.94|97.82|39.22|67.11|43.49|
> |Coronal|68.77|73.26|96.99|39.35|66.98|43.75|
> |Sagittal|67.21|80.21|91.41|39.42|65.66|45.31|
> |IILP (Ours)|**68.94**|**86.46**|**98.03**|**41.04**|67.31|**55.04**|
>
> &nbsp;
>
> > [W6] Lack clarity of certain experimental details.
>
> [A6] We respectfully clarify that the dataset used in Table 5 is explicitly stated in the footnote following line 579, indicating that the experiments were conducted on the TUEV dataset. Additionally, this footnote includes the rationale behind the dataset choice and detailed experimental setups. To enhance clarity and readability, we incorporated this dataset information directly into the caption of Table 5 in the revised manuscript. Regarding computational details, we have already discussed training resources and durations in lines 246–249 and 629–631. We have now included the fine-tuning duration details at line 631, indicating that the process typically takes from a few minutes to several hours, depending on the downstream dataset size.
>
> &nbsp;
>
> > [Q1] Exploration of activation functions for EEG foundation models.
>
> [A7] Thank you for raising this insightful question. We conducted additional experiments examining the impact of activation functions:
>
> |Activation Function / Datasets|TUEV|MentalArithmetic|Mumtaz2016|SEED-V|PhysioP300|BCIC-2A|
> |:-:|:-:|:-:|:-:|:-:|:-:|:-:|
> |ReLU|64.01|73.96|96.98|38.37|66.29|**47.31**|
> |GELU|64.02|73.61|**97.14**|38.69|64.38|46.35|
> |SwiGLU|**68.94**|**75.69**|97.07|**39.34**|**66.58**|44.01|
>
> We observed that different datasets exhibited distinct preferences: ReLU performed particularly well on BCIC2A, whereas GELU achieved competitive results on Mumtaz2016. Nonetheless, SwiGLU demonstrated the most consistent performance across multiple downstream EEG tasks, indicating its robustness for EEG data. Due to time constraints, these ablation experiments were conducted using models trained without the time-consuming pre-training stage. Inspired by your question, we plan to systematically explore and evaluate the impact of different activation functions during pre-training in future work.
>
> &nbsp;
>
> > [Q2] The appendix is provided separately.
>
> [A8] We apologize for any inconvenience caused by the separate submission of the appendix. When we submitted the main paper, we were still refining the presentation and structure of the appendix to ensure clarity and completeness of our extensive results from our downstream task experiments. Hence, we opted to include it as supplementary material .zip. We sincerely appreciate your patience and careful attention in reviewing the supplementary details.
>
> &nbsp;
>
> References:
>
> [1], [2] Attention is All You Need (NeurIPS 2017)
>
> [3] LaBraM (ICLR 2024)
>
> [4] Crossformer (ICLR 2023)
>
> ---
> **Thank you again for your valuable comments. We hope our responses have addressed your concerns and strengthened your impression of our work. We remain available throughout the rebuttal period to clarify any further questions you may have. Looking forward to hearing from you :-) !**

---

> ### Comment · Reviewer_L8YY · 2025-08-02
>
> The author's response addressed most of the reviewer's concerns, and the reviewer decided to maintain the positive rating while raising the clarity score to 3.

---

> ### Author Response · Authors · 2025-08-02
> **Thanks for your timely reply and the kind words**
>
> **Thank you for the positive rating and for informing us that our clarity score has improved. We're glad to hear that your concerns were adequately addressed during the rebuttal.**
>
> **We sincerely appreciate your professional and constructive feedback, which has substantially enhanced the readability and quality of our manuscript. All suggested revisions will be carefully incorporated into the final version.**
>
> **Should you have any additional questions or suggestions, please don't hesitate to reach out. We will remain actively engaged throughout the review process. Thank you once again for your valuable input.**

---

### Official Review · Reviewer_aqFB · 2025-06-30

**Clarity:** 3
**Significance:** 3
**Originality:** 2
**Rating:** 4
**Confidence:** 4

**Summary:**

Paper introduces a foundation model for EEG signals designed to handle the high variability across subjects, tasks, and recording setups. The model uses a transformer architecture with several novel components: 3D-aligned spatial encoding to handle different electrode montages, an amplitude-aware masking, and a MoE architecture to capture diverse temporal dynamics. Evaluation demonstrate good performance by fine-tuning the pre-trained model on several downstream tasks.

**Questions:**

Provide a more precise algorithmic description of how the masking interval [La, Ud] is defined based on the randomly sampled percentile g(i)? Is it a fixed-size window centered on that value, or does its width vary?


Method demonstrates strong fine-tuning performance, but a key promise of foundation models is few-shot or even zero-shot learning. Have you considered evaluating the model in a low-data regime (e.g., fine-tuning with 1% or 10% of the data) or in a few-shot setting?

3D electrode embedding is presented as a core spatial contribution, but its direct impact is not isolated in an ablation study. Could you compare its performance to a simpler learnable 1D channel embedding or something like in [1] to quantify the benefit of explicitly encoding physical coordinates?

**Ethical Concerns:**

["NO or VERY MINOR ethics concerns only"]

**Final Justification:**

Main concerns have been addressed.

**Limitations:**

Pre-training requires significant resources.

Model is evaluated on classification tasks.

May be discuss potential for misuse of powerful neural decoding models.

**Quality:**

3

**Strengths And Weaknesses:**

Amplitude-Aware Masking and Progressive Mixture-of-Experts (PMoE) combination is good to handle the unique characteristics of neural signals.

Evaluation across eight diverse datasets is another major strength, providing some evidence for the model's generalization capabilities.

Originality of some components is largely incremental, building heavily on existing work like Crossformer and MoE. Key implementation details, particularly for the AAMP masking strategy, are not sufficiently clear to ensure reproducibility. Furthermore, the performance gains on some datasets are marginal compared to baselines. Also how the masking is any different than what proposed earlier in [1], e.g., fixed cutout and random masking?

[1] Saeed, A., Grangier, D., Pietquin, O., & Zeghidour, N. (2021, June). Learning from heterogeneous EEG signals with differentiable channel reordering. In ICASSP 2021-2021 IEEE international conference on acoustics, speech and signal processing (ICASSP) (pp. 1255-1259). IEEE.

---

> ### Author Rebuttal · Authors · 2025-07-31
>
> **Dear reviewer aqFB:**
>
> **We appreciate your insightful comments. Below, we respond to each of your concerns.**
>
> ---
>
> > [W1] Originality of some components is largely incremental, building heavily on existing work like Crossformer and MoE. Key implementation details, particularly for the AAMP masking strategy, are not sufficiently clear to ensure reproducibility. Furthermore, the performance gains on some datasets are marginal compared to baselines. Also how the masking is any different than what proposed earlier in [1], e.g., fixed cutout and random masking?
>
> [A1] Our primary objective is to develop a more generalizable foundation model tailored for EEG data. While it is true that some components, such as Crossformer and Mixture-of-Experts (MoE), are inspired by existing work, it is noteworthy that these methods have not yet been demonstrated to be effective on the EEG foundation model. Our contribution lies in adapting, integrating, and validating the generalizability of these techniques for EEG representation learning to fill this critical gap. Rather than incremental reuse, we demonstrated that such architectures can significantly advance the state of the art in the EEG foundation model when used effectively with other components tailored for EEG.
>
> To ensure reproducibility, we have submitted our source code in the supplementary materials .zip file and on our anonymous GitHub. Moreover, we have improved writing on AAMP for better readability and clarity. Furthermore, a detailed description of the AAMP algorithm is provided in [A2].
>
> Regarding [1], the proposed CHARM addresses inconsistent input channels through a learnable channel-level masking approach, fundamentally different from our token-level AAMP method, which is tailored explicitly for representation learning in self-supervised foundational modeling. Specifically, Fixed Cutout masks out half the channels entirely, focusing solely on transfer learning with the remaining channels, whereas AAMP masks contiguous intervals of tokens based on amplitude thresholds for robust representation learning.
>
> We also notice COLA [2], which employs contrastive learning rather than masking-based self-supervised pretraining on large-scale audio data. These related efforts offer valuable complementary perspectives on enhancing time-series modeling. We have incorporated discussions of these references into our section on Related Work (Section A.1) to cover these important related works.
>
> [1] Learning from Heterogeneous EEG Signals with Differentiable Channel Reordering (ICASSP 2021)
>
> [2] Contrastive Learning of General-Purpose Audio Representations (ICASSP 2021)
>
> &nbsp;
>
> > [Q1] Provide a more precise algorithmic description of how the masking interval $[\mathcal{L}_d, \mathcal{U}_d]$ is defined based on the randomly sampled percentile $\xi_d^{(i)}$? Is it a fixed-size window centered on that value, or does its width vary?
>
> [A2] Given the randomly sampled percentile $\xi_d^{(i)}$ from the amplitude distribution of the EEG signal, we first identify its corresponding index in the sorted sequence $\textit{sorted}(x_d^{(i)})$, defined as $c = \lfloor \xi_d^{(i)} \cdot T \rfloor$. Subsequently, we define the amplitude interval $[\mathcal{L}_d, \mathcal{U}_d]$ symmetrically around this center point $c$, covering precisely $T \cdot \mathcal{P}$ points:
>
> $$\mathcal{L} _d = x _{(c - \lfloor T\mathcal{P}/2 \rfloor)}^{(i)}, \quad \mathcal{U} _d = x _{(c + \lceil T\mathcal{P}/2 \rceil)}^{(i)}$$
>
> Hence, while the masking proportion $\mathcal{P}$ remains fixed, the amplitude-based interval width $\mathcal{U}_d - \mathcal{L}_d$ dynamically varies depending on the local distribution of the amplitude values.
>
> &nbsp;
>
> > [Q2] Method demonstrates strong fine-tuning performance, but a key promise of foundation models is few-shot or even zero-shot learning. Have you considered evaluating the model in a low-data regime (e.g., fine-tuning with 1% or 10% of the data) or in a few-shot setting?
>
> [A3] While low-data scenarios have not been explored either in our original submission or in previous baseline works, we follow your suggestion and have conducted additional experiments with reduced training data:
>
> |Datasets|Data Used|Bal. Acc.|Cohen’s Kappa|Weighted F1|
> |:-:|:-:|-:|-:|-:|
> |TUEV|$1\\%$|64.16|72.44|88.08|
> ||$5\\%$|75.39|81.78|91.67|
> ||$10\\%$|80.30|88.64|95.75|
> ||$100\\%$|100.00|100.00|100.00|
> | Sleep-EDFx |$1\\%$|80.97|83.86|91.33|
> ||$5\\%$|86.01|79.82|90.41|
> ||$10\\%$|92.51|92.74|96.13|
> ||$100\\%$|100.00|100.00|100.00|
>
> \* *Metrics are shown as percentages relative to the 100% data baseline.*
>
> NeurIPT achieved notable classification accuracy, maintaining performance in the range of 80%-90% with merely 10% of the original data, and even with an extremely limited 1% of data, performance remains competitive at 64%-80%. These results further confirm the robustness of our pretraining strategy and underscore the model’s potential in few-shot and low-resource learning settings.
>
> &nbsp;
>
> > [Q3] 3D electrode embedding is presented as a core spatial contribution, but its direct impact is not isolated in an ablation study. Could you compare its performance to a simpler learnable 1D channel embedding or something like in [1] to quantify the benefit of explicitly encoding physical coordinates?
>
> [A4] We have now included a dedicated ablation study, comparing:
>
> - [3] trigonometric functions employed by vanilla Transformer,
> - [4] 1D learnable embeddings,
> - [5] 2D learnable embeddings introduced in LaBraM,
>
> |Datasets|Encoding Strategies|Bal. Acc.|Cohen’s Kappa|Weighted F1|
> |:-:|:-:|:-:|:-:|:-:|
> |TUEV|[3]|67.72|68.85|84.03|
> ||[4]|64.78|67.92|83.13|
> ||[5]|63.81|66.63|82.57|
> ||IILP (Ours)|**68.94**|**71.55**|**85.17**|
> |MentalArithmetic|[3]|74.65|71.84|83.80|
> ||[4]|71.53|74.23|83.17|
> ||[5]|71.52|64.81|80.83|
> ||IILP (Ours)|**75.69**|**74.37**|**86.83**|
> |Mumtaz2016|[3]|96.56|99.59|99.61|
> ||[4]|96.63|99.21|99.23|
> ||[5]|95.56|99.55|99.54|
> ||IILP (Ours)|**97.07**|**99.83**|**99.84**|
> |SEED-V|[3]|35.03|18.58|35.08|
> ||[4]|35.52|18.81|34.34|
> ||[5]|37.40|21.54|37.34|
> ||IILP (Ours)|**39.34**|**23.93**|**39.67**|
>
> Across all evaluated datasets, our method yields consistent and substantial improvements. This result confirms the tangible benefit of our spatial design while keeping the rest of the architecture unchanged, demonstrating that directly encoding the physical coordinates of electrodes provides stronger spatial information for the model.
>
> &nbsp;
>
> > [L1] Pre-training requires significant resources.
>
> [A5] We acknowledge that this is an inherent constraint shared by all contemporary foundation models. As detailed in [A3], NeurIPT has great potential in a low-data regime. Developing more efficient pre‑training strategies remains an important avenue for future work.
>
> &nbsp;
>
> > [L2] Model is evaluated on classification tasks.
>
> [A6] We acknowledge that our model is assessed on classification tasks, which are similar to all baseline methods. Expanding beyond classification, we consider it an essential direction for future research.
>
> &nbsp;
>
> > [L3] May be discuss potential for misuse of powerful neural decoding models.
>
> [A7] We have added a discussion on the potential misuse of powerful neural decoding models in Appendix G Discussion (Section G.3 Potential Misuse):
>
> As EEG foundation models become increasingly powerful for healthcare and brain–computer interface (BCI) applications, it is crucial to acknowledge their potential for misuse, for example, unauthorized cognitive surveillance or profiling. The ability to infer mental states or cognitive intent, if applied without proper oversight, raises profound ethical concerns. Addressing these risks will require the development of neurotechnology regulations, data protection policies, and ethics frameworks to ensure responsible use and safeguard individual neural privacy.
>
> &nbsp;
>
> References:
>
> [1] Learning from Heterogeneous EEG Signals with Differentiable Channel Reordering (ICASSP 2021)
>
> [2] Contrastive Learning of General-Purpose Audio Representations (ICASSP 2021)
>
> [3] [4] Attention is All You Need (NeurIPS 2017)
>
> [5] Large Brain Model for Learning Generic Representations with Tremendous EEG Data in BCI (ICLR 2024)
>
> ---
>
> **Thank you again for your valuable comments. We hope our responses have addressed your concerns and strengthened your impression of our work. We remain available throughout the rebuttal period to clarify any further questions you may have. Looking forward to hearing from you :-) !**

---

> ### Author Response · Authors · 2025-08-05
> **Thanks for your review and reconsideration**
>
> **We noticed that you have read and acknowledged our author rebuttal and have updated your assessment accordingly. We hope this indicates that your concerns have been adequately addressed.**
>
> **Should you have any further questions or suggestions, please don't hesitate to reach out. We remain actively engaged and are happy to respond throughout the extended discussion phase. Thank you once again for your valuable input.**

---

> > ### Comment · Reviewer_aqFB · 2025-08-05
> >
> > Thanks for the effort. My main concerns are addressed, I have updated the score.

---

> ### Author Response · Authors · 2025-08-05
> **Thanks for your timely reply and the kind words**
>
> **Thank you for your positive reply confirming that our responses have addressed your main concerns, and for raising your rating score.**
>
> **We sincerely appreciate your insightful and constructive feedback. As recommended, we will integrate all clarifications and additional experimental results into the final version. Thank you once again for the thorough review.**

---

### Official Review · Reviewer_qMmx · 2025-07-01

**Clarity:** 3
**Significance:** 3
**Originality:** 2
**Rating:** 4
**Confidence:** 5

**Summary:**

This paper introduces NEURIPT, a novel EEG foundation model for cross-task generalization, combining a Transformer architecture with an Amplitude-Aware Masked Pretraining (AAMP) module, a Progressive Mixture-of-Experts (PMoE) mechanism and an Intra-Inter Lobe Pooling (IILP) strategy. Evaluated across multiple datasets, NEURIPT outperforms existing methods in downstream tasks.

**Questions:**

The authors adopt the PMoE strategy with expert allocations [0, 0, 2, 4, 4, 6] across encoder layers (Table 3). However, an intuitively more gradual progression (e.g., [0, 2, 3, 4, 5, 6]) might better align with hierarchical feature learning in deeper layers. Could the authors clarify:

1. The design rationale for omitting experts in the first two layers and repeating identical expert counts in layers 4–5?

2. Whether empirical comparisons were conducted against alternative PMoE patterns (e.g., [0,2,3,4,5,6]), and if so, what trade-offs were observed in terms of performance?

**Ethical Concerns:**

["NO or VERY MINOR ethics concerns only"]

**Final Justification:**

All of my concerns have been adequately addressed in author's rebuttal.

**Limitations:**

yes

**Paper Formatting Concerns:**

1. Line 94 is missing a space before 'ranging'.

**Quality:**

3

**Strengths And Weaknesses:**

**Strengths**

1. This paper presents a well-structured and clearly written introduction that effectively contextualizes the challenges in EEG foundation model development.

2. The authors propose a novel amplitude-aware masked pretraining paradigm (AAMP), which departs from conventional random masking strategies to address signal heterogeneity.

3. The empirical evaluation demonstrates the model's competitiveness through comprehensive comparisons with diverse EEG foundation models (e.g., BIOT, EEGPT, CBraMod), achieving superior performance across eight downstream tasks.

4. The authors meticulously document training protocols, hyperparameters, and implementation details, ensuring strong reproducibility.

**Weaknesses**

1. The method section requires improved exposition, particularly regarding the symbolic notation. Ambiguous definitions of key components hinder readability and precise understanding of the model’s architecture.

2. The proposed AAMP lacks rigorous theoretical grounding. While empirically effective, the amplitude-based masking risks disrupting temporal continuity in EEG signals. As described in Eq. 4, masking intervals centered on random amplitude thresholds could lead to prolonged contiguous signal mask (contrary to the idealized example in Fig. 3b), raising concerns about the reliability of reconstruction and its alignment with EEG dynamics.

3. The ablation study in Appendix B (comparing AAMP to BERT-style random masking) is limited to a single downstream dataset (TUEV) with SVM-based evaluation. The observation that random masking achieves lower reconstruction loss (Fig. 7) but inferior downstream performance requires deeper analysis( the author's explanation fails to convince me). A more convincing validation would involve swapping only the masking strategy (keeping other components fixed) and evaluating across multiple tasks to isolate its impact.

4. While Fig. 2 highlights heterogeneous spectral patterns across downstream tasks, the model architecture does not explicitly incorporate frequency-band-specific mechanisms to exploit these differences. The IILP pooling strategy, though spatially informed, does not directly address the spectral heterogeneity emphasized in the introduction, limiting the interpretability of the proposed design choices.

5. The paper lacks comprehensive ablation studies to validate the contribution of each key component in NeurIPT (e.g., 3D positional encoding, PMoE, IILP).

6. The right panel of Fig. 5 (channel-wise analysis) suffers from insufficient font size and resolution, obscuring critical details in the visualization

---

> ### Author Rebuttal · Authors · 2025-07-31
>
> **Dear reviewer qMmx:**
>
> **We appreciate your insightful comments. Below, we respond to each of your concerns.**
>
> ---
>
> > [W1, Paper Formatting Concerns] The method section requires clearer symbolic notation and definitions.
>
> [A1] Thank you for requesting clarity on symbolic notation and definitions. We have now revised the manuscript, summarised as follows, for clarity and readability:
>
> - Removed the redundant definition $P_d$ at line 149 and explicitly clarified its relationship to the electrode channel $d$.
> - Revised the notation for positional embedding $PE$ in Eqs. 1 and 6 from boldface to italic, differentiating it from the bold matrix $E$, ensuring consistency with [1].
> - Simplified and rewrote the descriptions of the two sets presented at lines 179–180.
> - Inserted the term "labeled" at line 215 to explicitly correspond with the earlier description at line 144.
> - Refined the phrasing and terminology at lines 147, 159, 167, and 201.
>
> We hope the revision improves clarity. Should you have any additional suggestions, we would be grateful to receive them.
>
> &nbsp;
>
> > [W2, W3: AAMP] Explaination and ablation with random masking.
>
> [A2] We first clarify that the example in Fig. 3 is not just an idealized illustration but an actual sample from the TUEV dataset, reflecting the practical masking patterns produced by AAMP. Regarding temporal continuity concerns, the amplitude-based masking strategy is intentionally designed to produce longer contiguous masked segments with a similar amplitude range compared to random masking.
>
> Our amplitude-aware masking is a context-aware strategy that encourages the model to learn signal-specific characteristics and structural patterns, with amplitude serving as a proxy for signal energy. This approach reduces the risk of trivial reconstruction via simple interpolation methods (e.g., linear), which models might otherwise exploit. While prolonged masking may occur within individual channels, our model adopts a multivariate design, allowing it to reconstruct masked regions by leveraging complementary, unmasked information across channels. This preserves learning effectiveness even under dense intra-channel masking conditions.
>
> As highlighted in Fig. 7, although random masking achieves lower reconstruction loss, its inferior downstream performance strongly supports our central argument: random masking inherently promotes interpolation rather than encouraging deeper structural feature learning. To further substantiate this claim, we conducted additional experiments where only the masking strategy was swapped while keeping all other model components fixed:
>
> |Datasets|Masking Strategies|Bal. Acc.|Cohen’s Kappa|Weighted F1|
> |:-:|:-:|:-:|:-:|:-:|
> |TUEV|Random Masking|67.35|67.28|83.37|
> ||AAMP (Ours)|**67.61**|**69.70**|**84.28**|
> |MentalArithmetic|Random Masking|84.38|77.89|90.98|
> ||AAMP (Ours)|**86.46**|**78.27**|**91.11**|
> |Mumtaz2016|Random Masking|97.14|**99.84**|**99.84**|
> ||AAMP (Ours)|**98.03**|99.81|99.79|
> |SEED-V|Random Masking|33.89|17.07|33.82|
> ||AAMP (Ours)|**41.04**|**26.29**|**41.58**|
> |PhysioP300|Random Masking|65.64|28.99|72.75|
> ||AAMP (Ours)|**67.31**|**34.26**|**76.83**|
> |Sleep-EDFx|Random Masking|69.68|76.82|87.11|
> ||AAMP (Ours)|**70.47**|**77.57**|**87.39**|
>
> These results demonstrate consistent improvements of AAMP across multiple downstream datasets, demonstrating the effectiveness and generalization capability of our proposed masking strategy.
>
> &nbsp;
>
> > [W4] IILP doesn't directly address the heterogeneous spectral patterns mentioned in Fig. 2.
>
> [A3] We would like to clarify that, as stated in Lines 196-198 and 211-213, our Progressive Mixture-of-Experts (PMoE) architecture with a hierarchical structure in the backbone model is specifically designed to handle the heterogeneous and diverse information across various EEG signals, rather than the IILP pooling strategy. Figures 6 and 9-15 further substantiate this design, illustrating distinct activation patterns by different experts across multiple categories, thereby confirming that our specialized experts effectively capture the spectral heterogeneity present in EEG signals.
>
> &nbsp;
>
> > [W5] Ablation studies on 3D positional encoding, PMoE, IILP.
>
> [A4] We have expanded our ablation analyses across more downstream datasets. Specifically, comprehensive experiments for the component AAMP have been detailed in [A2]. Below, we provide comparisons focusing on the 3D positional encoding, PMoE, and the IILP strategies.
>
> |Datasets|Encoding Strategies|Bal. Acc.|Cohen’s Kappa|Weighted F1|
> |:-:|:-:|:-:|:-:|:-:|
> |TUEV|[1]|67.72|68.85|84.03|
> ||[2]|64.78|67.92|83.13|
> ||[3]|63.81|66.63|82.57|
> ||IILP (Ours)|**68.94**|**71.55**|**85.17**|
> |MentalArithmetic|[1]|74.65|71.84|83.80|
> ||[2]|71.53|74.23|83.17|
> ||[3]|71.52|64.81|80.83|
> ||IILP (Ours)|**75.69**|**74.37**|**86.83**|
> |Mumtaz2016|[1]|96.56|99.59|99.61|
> ||[2]|96.63|99.21|99.23|
> ||[3]|95.56|99.55|99.54|
> ||IILP (Ours)|**97.07**|**99.83**|**99.84**|
> |SEED-V|[1]|35.03|18.58|35.08|
> ||[2]|35.52|18.81|34.34|
> ||[3]|37.40|21.54|37.34|
> ||IILP (Ours)|**39.34**|**23.93**|**39.67**|
>
> alternative methods are included:
>
> - [1] trigonometric functions employed by vanilla Transformer,
> - [2] 1D learnable embeddings,
> - [3] 2D learnable embeddings introduced in LaBraM,
>
> The results above clearly demonstrate that our proposed 3D electrode embedding achieves consistently superior performance across multiple datasets, substantiating the notable advantage of explicitly leveraging physical electrode positions.
>
> | MoE Strategies / Datasets |   TUEV    | MentalArithmetic | Mumtaz2016 |  SEED-V   | PhysioP300 |  BCIC-2A  |
> | :-------------------: | :-------: | :--------------: | :--------: | :-------: | :--------: | :-------: |
> |      w/o Expert       |   65.83   |      72.92       |   93.41    |   39.14   |   64.53    |   44.44   |
> |        Uniform        |   65.91   |      70.49       |   95.00    |   39.33   |   65.66    |   41.93   |
> |       Shrinking       |   65.80   |      73.96       |   93.08    |   39.21   |   65.99    | **44.62** |
> |  Progressive (Ours)   | **68.94** |    **75.69**     | **97.07**  | **39.34** | **66.58**  |   44.01   |
>
> We have expanded Table 3 by evaluating different MoE strategies across multiple datasets, demonstrating the robustness of our PMoE approach.
>
> | Pooling Strategies / Datasets |   TUEV    | MentalArithmetic | Mumtaz2016 |  SEED-V   | PhysioP300 |  BCIC-2A  |
> | :-------------------: | :-------: | :--------------: | :--------: | :-------: | :--------: | :-------: |
> |      w/o Pooling      |   62.33   |      75.69       |   78.21    |   38.90   | **67.82**  |   45.14   |
> |     Mean Pooling      |   64.74   |      79.51       |   96.22    |   37.62   |   66.72    |   37.24   |
> |      Hemispheres      |   64.45   |      81.94       |   97.82    |   39.22   |   67.11    |   43.49   |
> |        Coronal        |   68.77   |      73.26       |   96.99    |   39.35   |   66.98    |   43.75   |
> |       Sagittal        |   67.21   |      80.21       |   91.41    |   39.42   |   65.66    |   45.31   |
> |      IILP (Ours)      | **68.94** |    **86.46**     | **98.03**  | **41.04** |   67.31    | **55.04** |
>
> We have expanded Table 4 by incorporating comparisons between IILP and alternative classification methods across multiple datasets, further demonstrating the consistent superiority of our approach.
>
> &nbsp;
>
> > [W6] Fig. 5 clarity.
>
> [A5] To enhance readability, we have revised the figure by increasing the font size and repositioning the channel names radially along the outer edge of the circular visualization. Additionally, we have updated the caption to emphasize that a consistent color scheme is applied in both left and right panels, enhancing the clarity of attention scores across brain regions.
>
> &nbsp;
>
> > [Q1, Q2] PMoE ablation and discussion.
>
> [A6] Indeed, assigning zero experts in the initial two encoder layers implies the direct use of standard FFNs rather than specialized experts. The rationale behind this choice is grounded in the principle that lower layers should acquire generalized representations beneficial across all tasks, while higher layers incrementally introduce specialization to handle more differentiated features. Regarding the repetition of expert counts in layers 4 and 5, it was not intentionally designed for specific benefits but resulted from a random initial allocation. Nevertheless, we have empirically compared our chosen PMoE pattern against alternative, more gradual configurations.
>
> | PMoE Strategies / Datasets |   TUEV    | MentalArithmetic | Mumtaz2016 |  SEED-V   | PhysioP300 |  BCIC-2A  |
> | :-------------------: | :-------: | :--------------: | :--------: | :-------: | :--------: | :-------: |
> |     [0,0,2,4,4,6]     | **68.94** |      75.69       |   97.07    |   39.34   |   66.58    | **44.01** |
> |     [0,0,2,3,4,5]     |   67.87   |    **76.39**     |   96.81    | **39.88** |   66.34    |   41.58   |
> |     [0,0,2,4,6,8]     |   66.41   |      75.69       |   96.57    |   39.11   | **67.70**  |   41.15   |
> |    [0,0,3,6,9,12]     |   66.08   |      74.83       | **97.15**  |   39.35   |   66.63    |   38.19   |
>
> These alternative patterns also yield competitive performance. This empirical observation suggested that the effectiveness of the PMoE approach stems from its inherent progressive strategy, rather than relying on any specific expert allocation, demonstrating robustness across diverse progressive configurations.
>
> &nbsp;
>
> References:
>
> [1] [2] Attention is All You Need (NeurIPS 2017)
>
> [3] Large Brain Model for Learning Generic Representations with Tremendous EEG Data in BCI (ICLR 2024)
>
> ---
>
> **Thank you again for your valuable comments. We hope our responses have addressed your concerns and strengthened your impression of our work. We remain available throughout the rebuttal period to clarify any further questions you may have. Looking forward to hearing from you :-) !**

---

> > ### Comment · Reviewer_qMmx · 2025-08-03
> >
> > Most of my concerns have been addressed. Regarding weak5, I would still encourage the authors to conduct ablation studies on the individual components of NeurIPT (e.g., w/o 3D positional encoding, w/o PMoE, w/o IILP, and w/ all components). While comparative experiments for each component can demonstrate their overall effectiveness, the specific contribution of each component to the performance of NeurIPT remains unclear. Such an ablation analysis would strengthen the interpretability and rigor of the proposed method.

---

> ### Author Response · Authors · 2025-08-03
> **We're glad to know most questions were addressed; We're now preparing the experiments**
>
> **We appreciate your confirmation that most of your concerns have been resolved, with one remaining issue requiring additional ablation. We agree that this ablation will further strengthen the interpretability and rigor of our work, and positively contribute to your overall assessment.**
>
> **We are currently actively working on recommended ablation analyses and will promptly share the detailed results once available. Thank you again for your constructive suggestion and continued patience!**

---

> ### Author Response · Authors · 2025-08-04
> **Response to ablation studies on each individual component**
>
> Thank you for requesting ablation analyses on each individual component, which we agree will enhance the interpretability and rigor of our work. Below, we provide a detailed investigation of each component in **NeurIPT**. (Kindly note that, due to time and resource constraints, the ablation results presented here are based on models trained from scratch, without the time-intensive pre-training stage. Nonetheless, the analyses remain valuable in illustrating the compounded impact of each component.) These findings will be incorporated within the additional page allowance in the final version.
>
> |   Components    |          |          | Datasets  |                      |                |            |                |
> | :-------------: | :------: | :------: | :-------: | :------------------: | :------------: | :--------: | :------------: |
> | **3D Positional Encoding** | **PMoE** | **IILP** | **TUEV**  | **MentalArithmetic** | **Mumtaz2016** | **SEED-V** | **BCIC-IV-2A** |
> |    &#10007;     | &#10007; | &#10007; |   51.80   |        73.36         |     91.83      |   37.82    |     32.64      |
> |    &#10004;     | &#10007; | &#10007; |   59.64   |        73.61         |     86.07      |   38.54    |     40.19      |
> |    &#10007;     | &#10004; | &#10007; |   52.79   |        74.65         |     85.58      |   37.82    |     33.59      |
> |    &#10007;     | &#10007; | &#10004; |   59.10   |        73.96         |     91.55      |   35.66    |     37.15      |
> |    &#10004;     | &#10004; | &#10007; |   62.33   |      **75.69**       |     78.21      |   38.90    |   **45.14**    |
> |    &#10004;     | &#10007; | &#10004; |   65.83   |        72.92         |     93.41      |   39.14    |     44.44      |
> |    &#10007;     | &#10004; | &#10004; |   67.72   |        74.65         |     96.56      |   35.03    |     37.59      |
> |    &#10004;     | &#10004; | &#10004; | **68.94** |      **75.69**       |   **97.07**    | **39.34**  |     44.01      |
>
> Empirically, the ablation analyses demonstrate that performance across datasets generally improves with the inclusion of each component, achieving peak accuracy when all components are combined. Notably, the IILP module significantly enhances results on the *TUEV* and *Mumtaz2016* datasets. This improvement is presumably because epilepsy and depression classification tasks require capturing distinctive regional signal variations across different brain areas.
>
> For the BCIC-IV-2A dataset, which primarily involves motor imagery tasks and was recorded using EEG channels located centrally with limited coverage of other brain lobes, we observe a performance decline when the 3D positional encoding component is excluded. This suggests that motor imagery tasks are particularly sensitive to spatial information, as further supported by our channel perturbation analysis in Fig. 5. Given the low-data scenario (BCIC-IV-2A includes only nine subjects), explicitly encoding physical spatial relationships becomes especially important. Moreover, the performance drop when only IILP is excluded may be attributed to the incomplete brain lobe coverage, which restricts the model’s ability to leverage lobe-specific information fully. Nevertheless, when IILP is incorporated during pretraining, it demonstrated strong generalization to BCIC-IV-2A, as shown in [A4] Table 4 of our previous response.

---

> > ### Comment · Reviewer_qMmx · 2025-08-04
> >
> > Thanks for the detailed response. All of my concerns have been adequately addressed. I hope these experiments can be added into the final version. I will raise my score to 4, good luck!

---

> ### Author Response · Authors · 2025-08-04
> **Thanks for your timely reply and the warm words!**
>
> **Thank you for your prompt and positive reply confirming that our responses have fully addressed your concerns, and for raising your rating score.**
>
> **We sincerely appreciate your professional and constructive feedback, which has greatly improved the clarity and quality of our manuscript. As recommended, we will integrate all clarifications and additional experimental results into the final version.**
>
> **Should you have any further comments or suggestions, please do not hesitate to reach out. We remain available throughout the revision process. Thank you once again for your valuable insights.**

---

### Author Response · Authors · 2025-08-06
**General Response and Summary of Revisions**

We sincerely thank all reviewers (qMmx, aqFB, L8YY, and LoU3) for their valuable and positive feedback, which has significantly enhanced the clarity of our work. We also appreciate that all reviewers confirmed their concerns were adequately addressed.

Moreover, we are grateful that all reviewers unanimously recognized that our paper is well-motivated and introduced several innovations that advance the current state of the EEG foundation model. In particular, the proposed Amplitude-Aware Masked Pretraining (AAMP), Progressive Mixture-of-Experts (PMoE), 3D electrode coordinate encoding, and Intra-Inter Lobe Pooling (IILP) were highlighted. Furthermore, we appreciate the reviewers’ acknowledgment of our comprehensive experiments across eight diverse EEG datasets and ablation analyses, which demonstrated the generalization capability and robustness of the proposed NeurIPT.

---

We have revised our manuscript based on the reviewers’ comments, incorporating both clarifications and additional experiments provided during the rebuttal and discussion phases. Below, we summarize the key modifications, following the narrative structure of the manuscript:

- L149: Removed the redundant definition $P_d$ and explicitly clarified its relationship to the electrode channel $d$. (Thanks to reviewer qMmx)
- L151: Revised the notation for positional embedding $PE$ in Eqs. 1 and 6 from boldface to italic, differentiating it from the bold matrix $E$, ensuring consistency with paper *"Attention is All You Need"*. (Thanks to reviewer qMmx)
- Fig. 5: Increased font size, repositioned channel names radially along the outer edge of the circular visualization, and updated the caption. (Thanks to reviewer qMmx)
- L179-180: Simplified and rewrote the descriptions of the two sets presented. (Thanks to reviewer qMmx, aqFB)
- L215: Inserted the term "labeled" to explicitly correspond with the earlier description. (Thanks to reviewer qMmx)
- L230: The reason for IILP gathering representations from all encoder layers. (Thanks to reviewer LoU3)
- L282: Extended Table 3 and Table 4 with rebuttal experiments and their corresponding analysis.  (Thanks to reviewer qMmx, aqFB, L8YY, LoU3)
- L282: Added Table 5 for ablation study on each individual component. (Thanks to reviewer qMmx)
- L566: Added experiments of activation functions and low-data scenario into Appendix B. (Thanks to reviewer aqFB, L8YY)
- L894: Added Appendix G.3 to discuss potential for misuse of powerful neural decoding models. (Thanks to reviewer aqFB)

---

Finally, we would like to express our gratitude once again to the reviewers for their detailed and thoughtful feedback. We deeply appreciate your invaluable contributions and remain available for further clarification should there be any additional questions or suggestions.

---

### Note · Authors · 2025-08-14

We sincerely appreciate the positive feedback from all reviewers and thank them for confirming that our responses have adequately addressed their concerns. We are especially grateful that three reviewers subsequently increased their scores accordingly.

Beyond addressing the reviewers’ valuable comments, we highlight the broader impact of our work on the EEG and Brain Computer Interface (BCI) communities: our proposed NeurIPT advances EEG foundation modeling by explicitly integrating core EEG-specific characteristics, yielding adaptable representations that support diverse task-specific scenarios and enable applications in clinical diagnostics, emotion recognition, and user-intent control. The study also opens promising avenues for future research, particularly the incorporation of additional neurophysiological factors such as brain connectivity.

Collectively, these contributions equip the BCI community with more generalizable, robust, and accessible EEG-based neural interfaces and, we believe, will catalyze practical, real-world deployments that ultimately enhance human well-being. We once again thank the reviewers for their engagement and recognition of our contributions, and we look forward to the continued growth of this research direction within the community.

---

### Decision · Program_Chairs · 2025-09-17

**Decision:**

Accept (poster)

**Comment:**

After the rebuttal, most of the concerns and questions were resolved, leading two reviewers to raise their scores by one point. As a result, all four reviewers gave a borderline accept rating. While it is true that none of the reviewers expressed particularly enthusiastic support, the performance across a wide range of benchmarks is solid, and no major concerns remain. On this basis, acceptance is considered a reasonable decision.